# Temperature variability projections remain uncertain after constraining them to best performing Large Ensembles of individual Climate Models

Laura Suarez-Gutierrez[1,2,3,6] ✉ & Nicola Maher [4,5,6]

Changes in temperature variability affect the frequency and intensity of extreme events, as well as the regional range of temperatures that ecosystems and society need to adapt to. While accurate projections of temperature variability are vital for understanding climate change and its impacts, they remain highly uncertain. We use rank-frequency analysis to evaluate the performance of eleven single model initial-condition large ensembles (SMILEs) against observations in the historical period, and use those that best represent historical regional variability to constrain projections of future temperature variability. Constrained projections from the best-performing SMILEs still show large uncertainties in the intensity and the sign of the variability change for large areas of the globe. Our results highlight poorly modelled regions where observed variability is not well represented such as large parts of Australia, South America, and Africa, particularly in their local summer season, underscoring the need for further modelling improvements over crucial regions. In these regions, the constrained projected change is typically larger than in the unconstrained ensemble, suggesting that in these regions, multi-model mean projections may underestimate future variability change.

To understand how global warming will affect the Earth's climate, society and ecosystems, we must not only understand the changes in the mean climate, but also whether and how climate variability will change[1]. Projecting changes in temperature variability is crucial for understanding potential future changes in extreme events, the range of temperature conditions that any given region could experience[1-4], and the impacts of these changes on ecosystem dynamics and habitats (e.g., refs. 5,6). This is because changes in extreme values in a distribution are affected by changes in the mean, its variance, and the skewness of the distribution[7-9]. For example, projected decreases in winter temperature variability in the northern hemisphere mid-

latitudes result in less frequent cold air outbreaks than expected by warming alone[10]. Furthermore, increases in summer temperature variability in Europe are expected to exacerbate the increases in heat extremes projected from global warming[2-4,11].

Although future changes in temperature variability can have important consequences, there are large uncertainties in the projections of how temperature variability will change under warming. The IPCC AR6 report flagged only medium confidence in changes in the extratropics and in the winter hemisphere mid-to-high latitudes, with low model agreement found everywhere else on the globe[12]. This uncertainty in the projected estimates of surface air temperature

[1]Institute for Atmospheric and Climate Science, ETH Zürich, Zurich, Switzerland. [2]Laboratoire des Sciences du Climat et de l'Environnement, Institut Pierre-Simon Laplace, Paris, France. [3]Meteorology and Air Quality Group, Wageningen University & Research, Wageningen, The Netherlands. [4]Research School of Earth Sciences, The Australian National University, Canberra, ACT, Australia. [5]ARC Centre of Excellence for the Weather of the 21st Century, The Australian National University, Canberra, ACT, Australia. [6]These authors contributed equally: Laura Suarez-Gutierrez, Nicola Maher. ✉e-mail: laura.suarez@wur.nl

variability applies not only to the intensity of the changes, but also to their sign, with low model agreement on the direction of the variability change over large areas[13].

While overall projections are uncertain, there is some consensus on how temperature variability will likely change over certain regions. Variability is projected to increase over the tropical land surface in CMIP5, single model initial-condition large ensembles (SMILEs), and 17-member Ensemble Simulations of Extreme Weather Events under Nonlinear Climate Change (ESSENCE) models due to drying soil in the southern hemisphere and increases in atmospheric variability in the northern hemisphere and thermal advection[1,14,15]. In Europe, projected reductions in winter variability are related to a lower land-sea temperature contrast[16] and a decrease in horizontal temperature gradients and therefore advection[17], while projected increases in summer variability[2] relate to changes in soil drought frequency[3,18], regions that lose summer snow cover[19], and changes in the surface heat balance[16]. In the mid-latitudes, in CMIP5 and ESSENCE, variability is projected to decrease due to decreases in meridional temperature gradients and sea ice loss[1,14], with 60N the latitude where the largest changes in fractional snow cover occur. The 60N latitude is also a region of decreasing variability, particularly in winter[20]. In the Arctic, a study using SMILEs shows that all models have increased temperature variability as long as sea ice exists, with variability decreasing once sea ice is lost[15]. This decrease can be constrained by choosing models that represent sea ice well, which causes the magnitude of the variability decrease to be larger than in unconstrained models[21].

Daily temperature variability projections follow a similar pattern to interannual variability, with a projected reduction in variability in northern mid-latitudes and projected increases in low latitudes. In the northern hemisphere summer, this reduction in the high latitudes is only in North America, the high Arctic, and Africa with increases elsewhere[4,22]. Although in general there are large regions where evidence on a broad scale shows agreement on the projected sign of the change, models can still have large differences in the magnitude of the change, demonstrating high uncertainty, particularly on an annual scale[23].

It remains, however, unclear whether climate models disagree on the direction and intensity of temperature variability changes because they do not adequately simulate present-day temperature variability, or whether they disagree due to intrinsic model differences in how the drivers of temperature variability change in a future climate. Therefore, a possible hypothesis is that the ability of models to project future temperature variability depends on their ability to simulate historical variability. If this were true, subsampling models based on their historical performance in capturing observed variability would yield reduced uncertainty in future projections. This means that by constraining projections to models that best match observations and agree on their historical variability representation, the range of future variability change projections across best-performing models would be reduced, and thus be less uncertain. In contrast, the alternative hypothesis is that different models' projections of future temperature variability depend not simply on past temperature variability, but rather on how the Earth system as a whole evolves over different regions in each model. If this were true, subsampling models based on their historical performance would not necessarily reduce future uncertainty, as long as those historically best-performing models still exhibit diverging behaviors in their representation of future temperature variability changes. In this study, we address these questions by, first evaluating in a multi-model super ensemble of SMILEs which models can adequately represent observed regional variability under current climate conditions. Second, we subsample the super ensemble to those adequate models, per season and region, and compute constrained future projections for regional temperature variability. Third, we compare the full ensemble with this subset of the ensemble to determine if the uncertainty in our projections is lowered by constraining them to only models that adequately capture historical variability.

There has been a suite of work that aims to understand the best way to weight and/or subsample climate model output (e.g., refs. 24–27). These studies typically consider two important factors: model performance and model independence[24]. For individual applications, subsampling for models that adequately represent observations can lead to a reduction of errors[28]. However, when considering future change, model skill has been found to only relate weakly to the projected change, highlighting that past model performance does not guarantee better future projections[29]. Climate models are also known to not be independent of each other[30,31], leading to multi-model means that are not created from independent samples. Studies such as[24] have shown that weighting by a combination of performance and independence can increase the skill of projections by up to 17%, highlighting the added value of using subsampling and constraining methods. The SMILEs used in our analysis are largely independent (as found by comparing the family trees in refs. 24,30) across CMIP5 and CMIP6 models (excluding GFDL-SPEAR-MED, which has not yet been included in previous work on this topic). Therefore, we limit the super-ensemble constraining to performance-only metrics. We note that some of the models used here reflect several generations of model development and are thus not independent, e.g., CanESM2 and CanESM5, CESM-LE and CESM2-LE, and MPI-GE5 and MPI-GE6, which likely share model features[32]. This study provides, to our knowledge, the first performance-constrained projections of temperature variability and its change globally.

Furthermore, previous work assessing variability changes typically confounds model differences and internal variability, by using a multi-model mean (e.g., ref. 1), or uses a single SMILE ignoring potential model differences[3], or uses several SMILEs, but does not consider model performance when assessing model uncertainty (e.g., ref. 13). SMILEs themselves are an invaluable tool for assessing projections of temperature variability[3,12,13,15,33]. Without SMILEs, long time periods are needed to sufficiently sample temperature variability. These long time periods, by necessity, encompass a range of warming levels above pre-industrial conditions, thus potentially experiencing changes in the variability itself. SMILEs, however, allow for temperature variability to be composited from all ensemble members at a given year (or period of years), therefore yielding time-varying (or global-warming level varying) estimates of temperature variability that are not confounded by potential underlying changes in variability.

SMILEs also allow for a more robust evaluation of model performance and comparison against observations, determining performance by assessing whether observations fall within the range of possible climate outcomes, now better-sampled by large ensembles of climate model simulations[33]. Thus, SMILEs allow us to evaluate climate model estimates of internal variability in the historical period more robustly than ever before. In this study, we make use of these advantages and build on previous work by employing and expanding the rank-frequency model evaluation framework of ref. 33 to assess how well models simulate historical summer and winter temperature variability. This rank-frequency evaluation framework, which resembles probabilistic forecast verification techniques in the climate prediction literature[34–36] is based on assessing whether observations occur across all ensemble ranks (i.e., the position an observation takes among the sorted ensemble members for a given time step) of an SMILE with comparable frequency. We then use the results of this region- and season-based evaluation to constrain projections of regional temperature variability for both seasons at five warming levels above pre-industrial conditions.

The aims of our study are threefold:
- Evaluate how well each SMILE captures historical temperature variability in both summer and winter seasons over individual regions.

- Assess model agreement in the projections of temperature variability across all SMILEs for both the sign of the change and the spread of the ensemble.
- Constrain the projections using the SMILEs that perform well over the historical period, and assess whether this provides an increase in model agreement for future projections.

## Results

### Temperature variability evaluation

We begin by evaluating model performance in capturing temperature variability over the historical record. For this, we apply an expanded version of the rank-frequency model evaluation framework presented in ref. 33 to detrended and non-detrended monthly mean temperature anomalies in 11 SMILEs compared to historical observations, for 9 Ocean and 24 Land regions in both the December, January, February (DJF) and June, July, August (JJA) seasons. We expand the rank-frequency evaluation framework in ref. 33 by incorporating two formally defined evaluation criteria (e.g., Fig. 1). This rank-frequency evaluation framework assesses model performance based on a simple principle: whether observations occur uniformly across the whole ensemble spread, or in other words, across all ensemble ranks. Traditionally, this has been assessed in the ensemble forecasting literature by evaluating rank histogram *flatness*[35,37]. However, the fact that we are evaluating free-running uninitialised simulations against a relatively short observational record means that the internal variability in the climate system may be insufficiently sampled on these time scales, and thus rank histograms may appear to be not flat, even for perfectly performing models.

To overcome this, we assess the flatness of the rank histogram using a perfect-model rank range test, which does not make any assumptions about the shape of the underlying distribution and takes into account both ensemble size and serial correlation. This perfect-model range is constructed by treating each ensemble member in each SMILE as if it were observations, and calculating the resulting spread of model rank histograms. This provides a distribution of rank histograms that reflect perfect-model behavior for the record length considered here, yet may indeed not be perfectly flat. This perfect-model rank range gives us a baseline of possible deviations from flatness for the observations rank histogram that could occur due to insufficient internal variability sampling or other factors. Therefore, as long as the observations rank histogram is within the perfect-model rank histogram range, non-uniformity in the observations rank frequencies could arise from insufficient variability sampling and does not necessarily prove that the considered model does not capture observations adequately. We use this perfect-model rank range to assess model performance for spatially averaged metrics. Thus, when the observational rank histogram is within the perfect-model rank histogram range (e.g., as for detrended and non-detrended L8 examples in Fig. 1a, e), the model captures the variability in regionally averaged temperatures adequately for the region and season considered (Criteria 1; see "Methods" for further details).

This first evaluation criteria based on spatially aggregated metrics (e.g., time series and histograms in Fig. 1a–h), is complemented by a second grid-cell based evaluation criteria (e.g., Fig. 1i, j), to account for biases at the grid-cell level that may be smoothed or compensated by spatial averages. This second test follows similar uniform rank frequency principles, but is simplified to allow for a computationally efficient grid-cell level performance assessment. Instead of assessing the full spectrum of ranks against a perfect-model range, we assess only the minimum and maximum ranks and central rank sections against fixed frequencies assumed to be a reflection of uniform rank frequency distribution. Thus, this test assesses whether anomalies at the grid-cell level show model biases by observations clustering around the center bounds of the simulated ensemble spread or falling

outside of its limits with too high frequencies (Criteria 2; see "Methods" for further details).

These two types of biases would equate to non-flat rank histograms. For variability overestimation biases, when observations cluster in the center of the ensemble spread, rank histograms would exhibit a mound or convex shape (e.g., detrended L12 (g) or both detrended and non-detrended L20 examples (d, h) in Fig. 1). For variability underestimation biases, when observations fall too frequently outside of both ends of the ensemble spread, rank histograms would exhibit a 'U' or concave shape (e.g., detrended L19 example in Fig. 1f). For observations falling too frequently outside of only one side of the ensemble spread, this could indicate an asymmetrical variability underestimation bias (isolated most clearly for detrended data) or a bias in the forced response (i.e., too strong or too weak warming signal in the model compared to observations, most clearly visible for non-detrended data). In this case, rank histograms would exhibit a slanted shape (e.g., non-detrended L19 (b) and L12 (c) examples in Fig. 1).

Therefore, the two evaluation criteria in this rank-frequency evaluation framework are as follows:

- Evaluation Criteria 1. Perfect-model rank range regional level performance: the observational rank histogram for regionally averaged temperatures must lie within the perfect-model rank range across all ranks (with a maximum 10% deviation).
- Evaluation Criteria 2. Threshold-based grid-cell level performance: at least 50% of the region's grid-cells must be unbiased, meaning, at the grid-cell level, (a) observations do not cluster excessively within the central percentiles of the ensemble (i.e., observations do not occur with more than 80% frequency within the central 75th percentile ensemble bounds, indicative of variability overestimation bias) and (b) fall outside the ensemble range too frequently (i.e., observations do not occur with more than 8% frequency outside of the ensemble spread, indicative of variability underestimation bias for detrended data).

Only when both criteria are fulfilled do we determine that the model in question offers an adequate performance for the region and season considered, and is thus selected to be part of the constrained ensemble for that particular region and season. To create our performance-constrained ensemble based on how well the models represent the variability in historical observations, and to minimize the effect of potential model biases in the representation of forced changes and warming trends on our evaluation, we base this constraint on an evaluation of detrended temperature data (e.g., Fig. 2a; right column in Fig. 1). However, for users interested in assessing model performance in capturing both temperature variability and forced changes, we also provide the full evaluation for non-detrended temperature data (e.g., Fig. 2b; left column in Fig. 1). We also provide the full results of this evaluation, including evaluation time series and rank histograms and results for Criteria 1 and the grid-cell level assessment maps for Criteria 2 for all 11 SMILEs separately, for the DJF and JJA seasons and for both detrended and non-detrended temperatures against GISTEMPv4 and ERSSTv5 observations, respectively for land and ocean regions, in the Supplementary Information Section 1.

The results summarized in the temperature variability Evaluation Matrices (Fig. 2) reflect both evaluation criteria. Numbers mark the percentage of grid cells in the region that exhibit an adequate, unbiased representation of observations. The color shading marks cases that fulfill Criteria 1 (with the observations rank histogram within the perfect-model range). For the regions and seasons that also fulfill Criteria 2 (at least 50% of the grid cells are non-biased) and are deemed to be adequately simulated by a given model both at regionally-aggregated and at the grid-cell level, the fields are highlighted in green and considered adequately captured and part of the constrained ensemble. When only Criteria 1 is fulfilled (the rank of the spatially

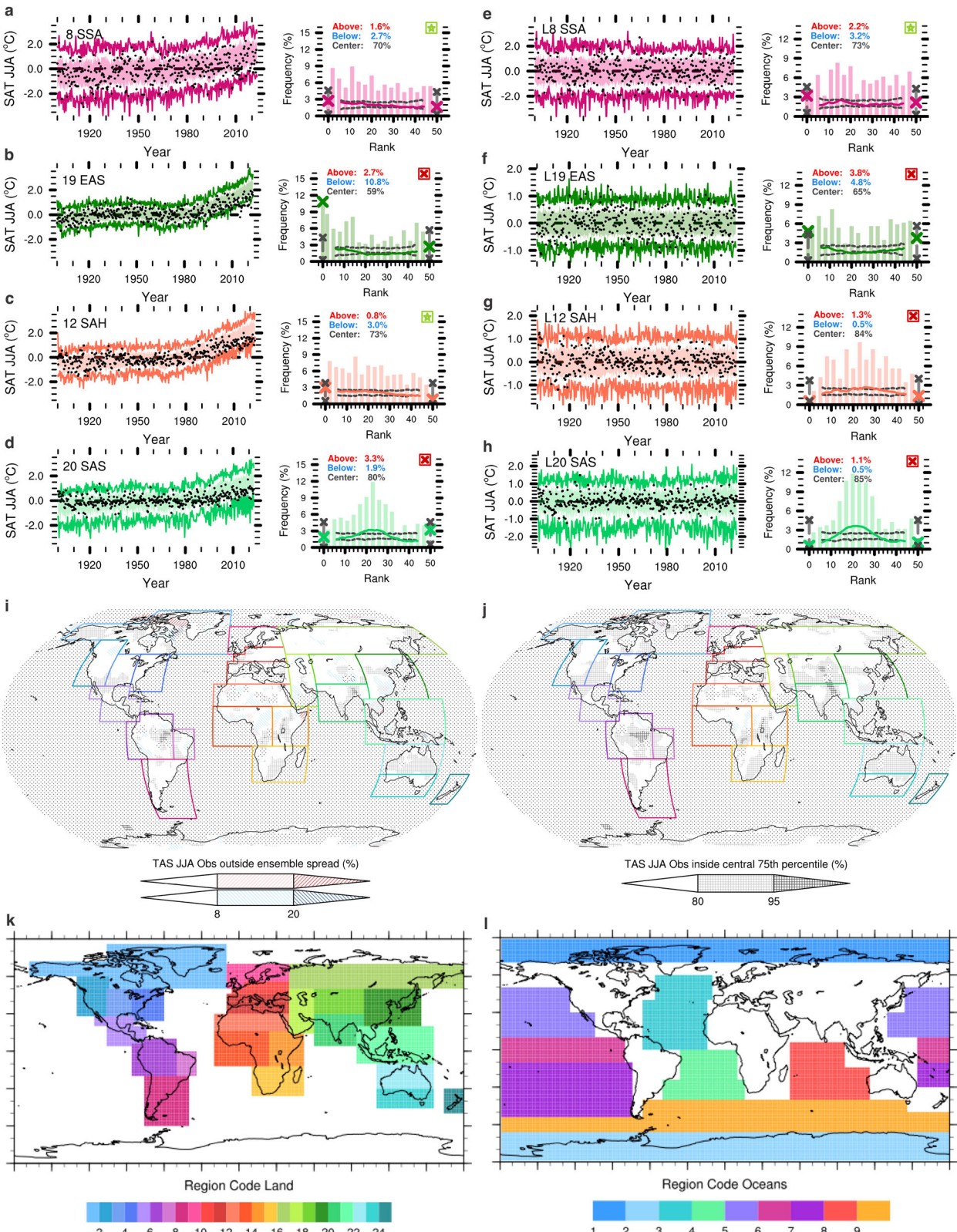

aggregated metric fits the perfect model range, but less than 50% of the grid cells are unbiased) the fields are highlighted in gray and considered inadequate.

Our evaluation shows that some regions are systematically not well represented (Fig. 2a) by all or most models across both seasons: e.g., Arctic (O1) and Antarctic (O2) Oceans, for which Criteria 1 is fulfilled, indicating adequate performance for spatially averaged

temperatures, but exhibit unbiased area fractions below 10% of the region's grid cells for most SMILEs. Other regions where most SMILEs offer an inadequate representation of temperature variability are the Southern Ocean (O9), the Indian Peninsula (L20), and Northern Australia (L22), as well as the Amazon basin (L6), Northern (L12) and Eastern Africa (L14), and Southern Australia (L23), all specifically in the local summer season. Our results also show that certain areas of the

**Fig. 1 | Example of the rank-frequency evaluation framework for June, July, August (JJA) temperatures in CanESM5 against GISTEMPv4 observations.** Time series and rank histograms for the evaluation of spatially averaged temperatures (Criteria 1) for non-detrended (**a**–**d**) and detrended data (**e**–**h**). Time series show ensemble maximum and minimum (lines) and central 75th percentile ensemble spread (shading) against observations (dots). Rank histograms show the observations rank frequency accumulated for 3-rank bins (bars), the running mean rank frequency over a centered $n/5$ rank window (lines; for 1 to $n-1$ ranks) and the absolute frequencies of rank 0 and $n$ (crosses), with $n$ the number of ensemble members, for observations (color) and perfect model rank range (gray). If observations rank frequency is within perfect-model range for all rank windows, Criteria 1 is met (highlighted by green star at the top right if Criteria 1 is met; if not, by a red cross). Percentages at the top left show the frequency of regionally averaged observations occurring above (red) or below (blue) ensemble limits, or clustering within the central 75th percentile range (gray), analogous to the grid-cell evaluation in Criteria 2. Center maps show the grid-cell evaluation of temperature variability (Criteria 2; **i** non-detrended temperatures, **j** detrended temperatures). Gray hatching (variability overestimation bias) shows observations clustering within the central 75th percentile with more than 80% frequency. Red and blue shading (variability underestimation bias) shows observations falling beyond the ensemble maximum or minimum, respectively, with more than 8% frequency. Dotted areas represent ocean grid cells or where observations are missing and therefore excluded. Bottom maps show region codes for land (**k**) and ocean (**l**) regions. Results for all 11 Single model initial-condition large ensembles (SMILEs) can be found in the Supplementary Information (SI).

world are systematically well captured by most models, especially in the local winter season, namely North America (L1–L4), Southern South America (L8), Europe and Northern Africa (L9–L11) and North and Central Asia (L16–L19). We also find that local winter seasons are generally simulated better by most models more often than local summer seasons. This may be because models struggle to simulate summer convective processes or land processes and land-atmosphere interactions more than they struggle to simulate large-scale winter circulation. Lastly, it is also worth noting that some of the regions that our evaluation highlights as systematically misrepresented in most models are regions with known widespread observational data quality issues, such as the Southern Ocean (O9) or Northern and Eastern Africa (L12, L14)[38–40]. This lack of sufficient temporal and spatial observational coverage could yield a potentially unreliable model performance assessment over these areas.

Our assessment also reveals some biases common across most models (Supplementary Information). Over land, most models show a marked overestimation of temperature variability over the tropics, especially over the Amazon basin, South Asia and Australia, both in DJF and JJA, and present for both detrended and non-detrended data. GFDL-SPEAR-MED and CESM-LE produce a substantially less biased temperature representation over these regions. Variability underestimation biases are relatively rare over land for most models, potentially due to our relatively conservative threshold for this bias to account for varying ensemble sizes (see "Methods" for further details). Variability underestimation biases are much more commonly found over the oceans, especially once the confounding effect of forced signal biases is removed by basing the evaluation on detrended data. These variability underestimation biases over the oceans occur most commonly over the Eastern Pacific and North Atlantic oceans and near sea-ice edges in both polar oceans. In contrast, variability over-estimation biases are also commonly found over the ocean for most models, especially over the Central Pacific and Southern Oceans, and both Polar Oceans, for which all models perform poorly across both seasons.

Strikingly, the Top 3 best performing models in our framework simulate 47 to 40 regions adequately, from a total of 66 regions across the 9 ocean regions and 24 land regions for the two seasons. The Top 3 best performing models in our framework in terms of highest number of adequately simulated regions (in brackets) are: CESM-LE (47), CESM2-LE (43), GFDL-SPEAR-MED (40). The Top 3 best performing models based on non-detrended data (Fig. 2b), meaning the models offering the best general model performance for both variability and forced changes simultaneously, are CESM-LE (33), CESM2-LE (28), and MPI-GE5 (26). SMILEs most affected by forced response biases, as in exhibiting the largest decreases in adequately simulated regions using non-detrended versus detrended data, are CSIRO-MK3.6 (with 33 adequately simulated regions with detrended data to 10 with non-detrended data), GFDL-SPEAR-MED (40 to 21 adequate regions), and ACCESS (28 to 11 adequate regions). This performance drop indicates that these models misrepresent forced changes over large areas; even

though some, such as GFDL-SPEAR-MED, may capture the variability in historical temperatures adequately.

The evaluation of non-detrended data (Fig. 2b and Supplementary Information Section 1) reveals generally worsened model performance over most ocean regions except the Northern Atlantic Ocean (O3) and over most generally poorly-model land areas, such as Central and South American (L4, L6, L7) and most of Asia and Oceania (L17–L24). This indicates that models present ample biases in their representation of forced changes over these areas, in addition to potential variability biases. However, in some limited cases, there may appear to be higher agreement between non-detrended data and observations than for detrended data (e.g., for L12 in JJA, as in Fig. 1 for CanESM5). This improved agreement could arise from the fact that misrepresented forcing signals can counteract commonly-found variability over-estimation biases, e.g., observations warming faster than the model ensemble and thus crossing the ensemble spread from its lower to upper sections in the historical period, yet remaining within the ensemble limits due to an overestimation of variability (as in L12 for Fig. 1 or in the idealized example in Fig. 1f in ref. 33). This is just an example of how forced response biases may confound variability assessments. To avoid these confounding effects, we base our constraint criteria exclusively on detrended data. Note, however, that, particularly for observations, detrending may not perfectly remove all forced changes (see "Methods" for further details) and may influence potential discrepancies between observed and simulated detrended data. Users that seek an assessment of which models best capture real world behavior as observed, including both variability and forced signal responses simultaneously, may again refer to this non-detrended evaluation (Supplementary Information Section 1 and Fig. 2b).

## Constrained vs. unconstrained temperature variability projections

The projections of temperature variability in each SMILE for all 5 warming levels and each region for DJF (Fig. 3) and JJA (Fig. 4) show that, typically, the multi-ensemble mean (MEM) exhibits the same sign of the change with increasing warming level in the full unconstrained ensemble as in the constrained ensemble (same sign of change as compared to the first bar which is variability at 1 degree of warming). Beyond the sign of the change, the absolute magnitude of the MEM temperature variability is qualitatively lower or similar in the constrained ensemble as in the unconstrained ensemble (see difference between colored and black solid lines). The exceptions where variability is larger in the constrained ensemble are Central North America (L3), Northern Asia (L16), Central Asia + Tibetan Plateau (L18), Eastern Asia (L19), South Pacific (O7), Indian (O8) in DJF and North Atlantic (O3), South Pacific (O7) in JJA. These results indicate that the MEM is generally a good representation of the sign of the change, also in the full unconstrained ensemble, but that the magnitude of the variability itself tends to be overestimated in the full ensemble as compared to the constrained ensemble. Additionally, while the constraint does not

**a.** Detrended Temperature Evaluation

| | ACCESS | | CanESM2 | | CanESM5 | | CESM-LE | | CESM2-LE | | CSIRO-MK3.6 | | GFDL-ESM2M | | GFDL-SPEAR-MED | | MIROC6 | | MPI-GE5 | | MPI-GE6 | | TOTAL | | |
|---|---|---|---|---|---|---|---|---|---|---|---|---|---|---|---|---|---|---|---|---|---|---|---|---|---|
| | DJF | JJA | DJF | JJA | DJF | JJA | DJF | JJA | DJF | JJA | DJF | JJA | DJF | JJA | DJF | JJA | DJF | JJA | DJF | JJA | DJF | JJA | DJF | JJA | |
| O1 | 4,9 | 8,4 | 2,1 | 9,3 | 4,9 | 5,0 | 6,9 | 2,3 | 2,9 | 7,5 | 3,2 | 17,9 | 5,9 | 5,5 | 1,9 | 5,9 | 6,8 | 8,4 | 8,0 | 11,4 | 6,4 | 10,4 | 0 | 0 | O1 |
| O2 | 5,6 | 8,2 | 15,7 | 3,7 | 5,5 | 13,0 | 26,1 | 9,1 | 26,9 | 16,2 | 4,8 | 4,8 | 17,9 | 7,8 | | | 13,4 | 2,5 | 7,8 | 16,3 | 8,3 | 13,7 | 0 | 0 | O2 |
| O3 | 55,6 | 72,4 | 52,7 | 64,5 | 61,6 | 56,6 | 84,0 | 89,7 | 80,1 | 74,5 | 71,3 | 69,9 | 65,9 | 56,9 | 69,7 | 70,1 | 64,1 | 55,8 | 34,3 | 45,1 | 43,1 | 48,0 | 3 | 5 | O3 |
| O4 | 84,2 | 77,3 | 83,8 | 73,1 | 81,3 | 78,3 | 61,6 | 89,6 | 89,5 | 85,7 | 84,3 | 80,8 | 86,3 | 54,8 | 89,0 | 74,1 | 81,8 | 79,8 | 64,0 | 47,8 | 68,2 | 61,2 | 8 | 9 | O4 |
| O5 | 72,2 | 70,0 | 76,1 | 58,7 | 69,9 | 57,2 | 73,4 | 65,6 | 56,8 | 51,0 | 53,6 | 43,1 | 53,9 | 64,3 | 58,4 | 62,8 | 55,0 | 60,2 | 33,9 | 58,3 | 42,5 | 59,5 | 6 | 8 | O5 |
| O6 | 63,6 | 60,4 | 48,9 | 49,9 | 75,8 | 62,5 | 57,4 | 69,5 | 8,4 | 8,4 | 50,7 | 47,4 | 46,7 | 22,7 | 84,1 | 73,8 | 55,8 | 9,2 | 67,7 | 32,6 | 63,0 | 41,0 | 6 | 4 | O6 |
| O7 | 61,6 | 54,1 | 67,1 | 69,5 | 59,1 | 70,5 | 79,6 | 68,7 | 30,9 | 56,3 | 60,2 | 51,8 | 48,1 | 66,0 | 61,2 | 73,1 | 46,6 | 52,3 | 45,6 | 43,4 | 45,8 | 51,4 | 4 | 7 | O7 |
| O8 | 87,9 | 90,9 | 55,6 | 64,1 | 61,9 | 72,5 | 57,9 | 72,9 | 67,1 | 69,0 | 55,7 | 60,6 | 54,5 | 46,4 | 69,8 | 78,3 | 51,1 | 55,9 | 34,6 | 46,8 | 49,1 | 61,6 | 8 | 8 | O8 |
| O9 | 32,2 | 50,7 | 48,8 | 45,3 | 51,8 | 40,9 | 55,1 | 52,8 | 38,5 | 65,2 | 52,4 | 43,8 | 34,5 | 46,6 | 38,9 | 41,4 | 50,6 | 47,9 | 31,4 | 33,2 | 37,7 | 40,9 | 1 | 1 | O9 |
| L1 | 82,0 | 42,9 | 50,2 | 35,0 | 58,5 | 38,1 | 68,5 | 61,5 | 89,3 | 26,0 | 77,8 | 46,9 | 62,2 | 56,3 | 86,6 | 58,6 | 78,0 | 60,2 | 80,6 | 61,8 | 82,9 | 58,3 | 11 | 4 | L1 |
| L2 | 60,5 | 37,4 | 63,8 | 32,4 | 45,8 | 67,3 | 60,0 | 81,2 | 82,6 | 33,3 | 70,4 | 92,0 | 74,6 | 24,4 | 92,6 | 79,1 | 77,0 | 56,7 | 67,4 | 59,7 | 76,2 | 61,5 | 8 | 6 | L2 |
| L3 | 75,8 | 2,1 | 72,9 | 27,6 | 61,7 | 29,3 | 65,4 | 75,4 | 100,0 | 71,5 | 95,0 | 42,8 | 71,9 | 43,1 | 100,0 | 62,5 | 92,1 | 59,0 | 54,0 | 56,3 | 71,3 | 58,9 | 9 | 6 | L3 |
| L4 | 94,2 | 52,0 | 81,6 | 19,7 | 82,2 | 90,2 | 73,1 | 77,8 | 98,5 | 49,6 | 78,7 | 31,5 | 95,7 | 43,1 | 98,5 | 90,3 | 89,0 | 88,6 | 57,5 | 97,4 | 74,4 | 94,6 | 11 | 5 | L4 |
| L5 | 71,9 | 35,3 | 35,8 | 40,5 | 72,1 | 75,0 | 60,5 | 53,6 | 90,3 | 56,1 | 51,3 | 41,5 | 46,5 | 40,3 | 72,2 | 58,3 | 54,3 | 6,5 | 56,9 | 46,6 | 51,8 | 46,8 | 6 | 3 | L5 |
| L6 | 53,6 | 36,3 | 15,6 | 21,0 | 23,8 | 38,8 | 66,7 | 75,6 | 34,3 | 58,3 | 32,1 | 46,2 | 24,2 | 17,4 | 65,2 | 69,4 | 14,6 | 14,4 | 57,8 | 42,6 | 58,3 | 39,0 | 3 | 1 | L6 |
| L7 | 23,9 | 38,2 | 9,5 | 17,5 | 61,5 | 40,0 | 71,3 | 76,4 | 43,1 | 65,2 | 28,4 | 77,8 | 39,9 | 9,6 | 72,8 | 70,0 | 15,9 | 6,1 | 64,7 | 79,1 | 55,1 | 71,3 | 4 | 5 | L7 |
| L8 | 63,3 | 78,7 | 32,3 | 69,3 | 65,3 | 69,8 | 83,5 | 80,5 | 50,8 | 88,8 | 56,9 | 79,8 | 34,5 | 56,3 | 66,7 | 86,6 | 32,6 | 79,8 | 52,4 | 57,4 | 56,0 | 61,2 | 6 | 10 | L8 |
| L9 | 78,4 | 93,7 | 64,9 | 33,0 | 77,7 | 86,2 | 51,8 | 92,8 | 93,7 | 79,2 | 27,2 | 88,5 | 98,8 | 97,2 | 85,1 | 97,9 | 92,1 | 83,3 | 73,7 | 100,0 | 77,0 | 98,0 | 8 | 9 | L9 |
| L10 | 81,9 | 33,8 | 89,4 | 11,9 | 89,2 | 85,8 | 83,0 | 93,2 | 99,1 | 56,0 | 39,3 | 13,5 | 98,9 | 44,2 | 97,8 | 77,2 | 65,6 | 69,6 | 53,8 | 79,7 | 63,7 | 80,7 | 8 | 6 | L10 |
| L11 | 81,9 | 63,2 | 72,6 | 40,0 | 83,1 | 69,2 | 89,9 | 62,6 | 97,6 | 68,1 | 67,9 | 53,1 | 89,9 | 27,3 | 88,5 | 77,5 | 79,8 | 47,8 | 89,3 | 38,9 | 88,5 | 42,8 | 9 | 6 | L11 |
| L12 | 90,9 | 74,7 | 92,4 | 65,2 | 95,5 | 67,5 | 93,1 | 65,9 | 91,9 | 53,6 | 84,6 | 49,4 | 95,0 | 18,2 | 94,3 | 77,1 | 88,1 | 59,3 | 94,9 | 42,6 | 93,3 | 47,7 | 9 | 2 | L12 |
| L13 | 65,9 | 63,5 | 73,1 | 52,6 | 76,1 | 54,8 | 69,4 | 78,7 | 78,8 | 59,2 | 56,0 | 74,3 | 61,6 | 39,7 | 79,2 | 69,6 | 60,7 | 38,6 | 55,3 | 25,2 | 63,3 | 29,3 | 7 | 4 | L13 |
| L14 | 79,6 | 55,1 | 52,3 | 42,7 | 55,4 | 44,1 | 76,8 | 46,9 | 83,0 | 51,0 | 61,9 | 59,2 | 56,9 | 24,5 | 59,7 | 48,3 | 53,0 | 26,6 | 56,3 | 19,8 | 61,7 | 20,4 | 7 | 1 | L14 |
| L15 | 45,4 | 49,8 | 41,4 | 37,5 | 54,0 | 58,0 | 72,3 | 85,3 | 59,1 | 65,3 | 77,8 | 76,2 | 59,8 | 34,6 | 56,5 | 48,4 | 23,0 | 64,5 | 49,7 | 37,3 | 42,8 | 30,8 | 3 | 4 | L15 |
| L16 | 74,6 | 84,5 | 81,8 | 29,8 | 90,1 | 84,5 | 59,8 | 76,7 | 97,0 | 16,4 | 67,3 | 67,0 | 97,4 | 62,5 | 86,7 | 78,6 | 84,7 | 84,8 | 93,7 | 56,7 | 96,0 | 59,5 | 8 | 8 | L16 |
| L17 | 78,3 | 74,4 | 75,5 | 56,7 | 73,0 | 62,1 | 80,0 | 59,0 | 93,1 | 71,3 | 40,2 | 57,2 | 67,9 | 27,3 | 74,2 | 76,2 | 82,7 | 46,9 | 87,1 | 34,6 | 88,6 | 37,4 | 7 | 7 | L17 |
| L18 | 85,3 | 70,2 | 77,5 | 40,3 | 60,7 | 37,0 | 55,4 | 80,4 | 82,2 | 79,7 | 62,6 | 95,2 | 76,4 | 46,1 | 75,6 | 73,2 | 65,4 | 24,5 | 81,6 | 32,1 | 82,8 | 33,2 | 7 | 4 | L18 |
| L19 | 74,8 | 48,9 | 42,5 | 49,1 | 52,3 | 65,3 | 82,6 | 86,2 | 91,7 | 63,1 | 83,4 | 57,8 | 81,5 | 48,4 | 95,0 | 76,6 | 52,9 | 65,2 | 64,0 | 62,0 | 71,3 | 55,5 | 8 | 9 | L19 |
| L20 | 40,5 | 51,3 | 19,1 | 20,1 | 11,5 | 45,6 | 33,2 | 40,0 | 45,2 | 20,7 | 56,9 | 49,0 | 18,0 | 15,6 | 39,9 | 48,0 | 35,9 | 6,0 | 16,7 | 24,3 | 20,1 | 18,2 | 1 | 0 | L20 |
| L21 | 47,8 | 55,1 | 43,7 | 49,3 | 33,8 | 29,6 | 41,7 | 75,0 | 68,0 | 73,2 | 48,0 | 69,6 | 41,1 | 42,5 | 54,6 | 52,0 | 34,8 | 29,9 | 58,5 | 61,7 | 51,4 | 52,2 | 4 | 5 | L21 |
| L22 | 23,2 | 39,7 | 4,5 | 33,1 | 14,5 | 20,4 | 54,0 | 90,4 | 66,4 | 45,2 | 48,6 | 73,1 | 12,0 | 35,0 | 12,9 | 63,4 | 6,0 | 71,4 | 9,8 | 37,5 | 9,0 | 46,2 | 1 | 2 | L22 |
| L23 | 37,9 | 91,8 | 37,8 | 49,4 | 48,1 | 68,1 | 77,6 | 96,6 | 74,7 | 89,3 | 75,3 | 81,4 | 46,4 | 79,4 | 42,8 | 66,5 | 22,3 | 87,9 | 11,8 | 59,1 | 22,0 | 55,5 | 2 | 8 | L23 |
| L24 | 93,0 | 76,3 | 100,0 | 68,3 | 92,7 | 29,4 | 92,2 | 91,6 | 92,7 | 100,0 | 78,4 | 30,6 | 61,2 | 93,0 | 85,7 | 93,0 | 100,0 | 100,0 | 67,5 | 100,0 | 69,0 | 100,0 | 9 | 7 | L24 |
| TOTAL | 28 | | 21 | | 32 | | 47 | | 43 | | 33 | | 19 | | 40 | | 26 | | 33 | | 31 | | | | |
| T_Season | 15 | 13 | 13 | 8 | 19 | 13 | 21 | 26 | 23 | 20 | 16 | 17 | 10 | 9 | 21 | 19 | 16 | 10 | 20 | 13 | 18 | 13 | | | |
| T_Ocean | 4 | 4 | 4 | 4 | 2 | 3 | 5 | 7 | 3 | 5 | 6 | 4 | 2 | 4 | 5 | 5 | 2 | 2 | 1 | 1 | 2 | 3 | | | |
| T_Land | 11 | 9 | 9 | 4 | 17 | 10 | 16 | 19 | 20 | 15 | 10 | 13 | 8 | 5 | 16 | 14 | 14 | 8 | 19 | 12 | 16 | 10 | | | |

**b.** Non-detrended Temperature Evaluation

| | ACCESS | | CanESM2 | | CanESM5 | | CESM-LE | | CESM2-LE | | CSIRO-MK3.6 | | GFDL-ESM2M | | GFDL-SPEAR-MED | | MIROC6 | | MPI-GE5 | | MPI-GE6 | | TOTAL | | |
|---|---|---|---|---|---|---|---|---|---|---|---|---|---|---|---|---|---|---|---|---|---|---|---|---|---|
| | DJF | JJA | DJF | JJA | DJF | JJA | DJF | JJA | DJF | JJA | DJF | JJA | DJF | JJA | DJF | JJA | DJF | JJA | DJF | JJA | DJF | JJA | DJF | JJA | |
| O1 | 21,2 | 13,1 | 3,8 | 10,5 | 15,4 | 19,1 | 8,4 | 6,8 | 20,8 | 33,0 | 32,9 | 35,7 | 14,8 | 27,1 | 13,1 | 21,5 | 3,0 | 22,4 | 13,0 | 34,0 | 9,8 | 18,0 | 0 | 0 | O1 |
| O2 | 7,3 | 7,1 | 12,9 | 3,3 | 10,2 | 4,6 | 5,3 | 6,9 | 17,4 | 12,7 | 57,8 | 13,5 | 7,9 | 4,1 | 18,6 | 9,3 | 19,5 | 2,4 | 11,3 | 6,9 | 13,6 | 6,1 | 0 | 0 | O2 |
| O3 | 41,7 | 63,6 | 32,1 | 55,1 | 49,5 | 63,9 | 76,1 | 82,8 | 87,9 | 78,4 | 60,9 | 71,8 | 43,8 | 65,6 | 63,1 | 74,2 | 56,0 | 56,6 | 40,8 | 53,2 | 47,3 | 51,7 | 4 | 3 | O3 |
| O4 | 66,3 | 64,6 | 40,3 | 45,7 | 57,0 | 66,7 | 37,2 | 49,8 | 87,7 | 82,7 | 57,1 | 43,5 | 55,2 | 61,1 | 86,2 | 67,5 | 56,5 | 70,7 | 63,2 | 58,2 | 68,5 | 66,0 | 0 | 0 | O4 |
| O5 | 78,1 | 73,5 | 60,3 | 54,9 | 68,3 | 56,1 | 75,9 | 66,5 | 62,4 | 62,0 | 52,2 | 43,9 | 68,6 | 78,4 | 66,0 | 63,4 | 78,2 | 61,2 | 45,4 | 64,0 | 56,8 | 66,0 | 2 | 3 | O5 |
| O6 | 68,9 | 75,8 | 51,4 | 49,8 | 49,1 | 75,5 | 66,0 | 74,7 | 12,1 | 13,3 | 44,9 | 56,5 | 58,8 | 42,9 | 77,8 | 75,9 | 70,9 | 14,8 | 66,1 | 41,4 | 62,7 | 47,2 | 6 | 2 | O6 |
| O7 | 71,7 | 59,1 | 66,6 | 62,8 | 57,5 | 60,7 | 81,5 | 74,0 | 36,2 | 59,6 | 68,0 | 48,1 | 61,5 | 68,4 | 70,4 | 74,5 | 71,7 | 60,1 | 49,1 | 58,8 | 53,1 | 60,9 | 2 | 2 | O7 |
| O8 | 92,8 | 72,7 | 62,9 | 61,0 | 81,9 | 78,8 | 59,4 | 73,1 | 70,1 | 72,5 | 66,9 | 51,7 | 67,2 | 56,2 | 77,0 | 81,8 | 68,3 | 60,9 | 36,9 | 61,7 | 53,7 | 72,2 | 5 | 1 | O8 |
| O9 | 40,7 | 39,8 | 54,3 | 31,0 | 41,7 | 31,4 | 53,5 | 54,1 | 52,5 | 51,1 | 43,0 | 26,8 | 43,8 | 33,3 | 48,0 | 34,6 | 38,3 | 39,4 | 43,6 | 47,4 | 51,3 | 47,6 | 0 | 0 | O9 |
| L1 | 80,1 | 49,4 | 57,3 | 44,9 | 53,8 | 48,4 | 72,9 | 63,7 | 88,6 | 35,3 | 79,2 | 44,0 | 65,7 | 54,3 | 83,7 | 51,1 | 81,4 | 60,9 | 81,1 | 59,3 | 82,1 | 57,3 | 8 | 3 | L1 |
| L2 | 69,9 | 66,8 | 68,1 | 74,1 | 52,9 | 74,6 | 63,6 | 79,8 | 86,4 | 62,2 | 72,1 | 88,1 | 85,2 | 37,4 | 94,1 | 77,4 | 74,5 | 54,3 | 67,3 | 59,0 | 78,6 | 60,8 | 7 | 3 | L2 |
| L3 | 74,6 | 2,9 | 79,7 | 63,8 | 66,4 | 66,8 | 67,9 | 54,3 | 100,0 | 95,6 | 84,9 | 58,2 | 81,3 | 50,0 | 98,9 | 55,9 | 89,3 | 48,3 | 62,5 | 57,1 | 77,3 | 53,9 | 7 | 3 | L3 |
| L4 | 93,1 | 63,2 | 80,7 | 35,0 | 80,7 | 94,8 | 70,6 | 65,7 | 97,5 | 61,5 | 68,1 | 42,8 | 88,0 | 73,0 | 97,1 | 67,8 | 88,5 | 83,8 | 59,9 | 90,4 | 80,3 | 78,3 | 5 | 2 | L4 |
| L5 | 72,2 | 38,6 | 33,9 | 54,9 | 68,9 | 58,3 | 57,3 | 45,4 | 79,1 | 56,3 | 51,5 | 55,9 | 46,1 | 45,3 | 67,4 | 51,6 | 56,1 | 13,3 | 51,9 | 53,4 | 43,6 | 43,6 | 5 | 4 | L5 |
| L6 | 47,1 | 38,6 | 31,2 | 35,2 | 34,8 | 52,1 | 57,6 | 54,4 | 38,5 | 71,4 | 24,9 | 40,6 | 29,0 | 37,9 | 45,7 | 59,3 | 25,1 | 34,8 | 62,8 | 52,2 | 52,2 | 49,8 | 1 | 2 | L6 |
| L7 | 30,2 | 46,2 | 23,8 | 31,8 | 53,6 | 44,5 | 31,4 | 22,1 | 65,4 | 63,6 | 33,0 | 36,2 | 28,7 | 28,5 | 56,9 | 53,8 | 31,5 | 39,2 | 58,5 | 71,2 | 58,5 | 71,2 | 3 | 0 | L7 |
| L8 | 61,2 | 71,9 | 52,1 | 80,8 | 75,5 | 77,8 | 74,3 | 80,9 | 63,4 | 87,1 | 54,2 | 68,1 | 36,0 | 69,1 | 63,3 | 86,2 | 35,2 | 75,8 | 55,5 | 64,2 | 55,2 | 70,1 | 5 | 4 | L8 |
| L9 | 65,4 | 91,6 | 62,9 | 53,6 | 69,5 | 90,9 | 52,5 | 96,1 | 86,5 | 83,1 | 35,4 | 83,6 | 96,8 | 91,8 | 94,1 | 75,5 | 87,0 | 85,7 | 86,4 | 100,0 | 87,1 | 94,0 | 6 | 7 | L9 |
| L10 | 73,5 | 47,2 | 82,6 | 11,9 | 87,1 | 89,2 | 68,4 | 93,2 | 96,2 | 67,5 | 56,0 | 23,7 | 99,0 | 58,7 | 98,9 | 77,3 | 73,0 | 73,7 | 74,6 | 76,4 | 85,1 | 80,8 | 7 | 6 | L10 |
| L11 | 63,9 | 61,5 | 71,1 | 38,6 | 70,0 | 74,7 | 92,1 | 65,3 | 96,1 | 68,0 | 66,2 | 56,8 | 92,4 | 33,9 | 83,0 | 73,7 | 79,7 | 54,3 | 91,8 | 46,9 | 87,2 | 47,6 | 7 | 4 | L11 |
| L12 | 71,6 | 54,0 | 64,1 | 70,9 | 70,8 | 63,1 | 83,4 | 73,0 | 86,9 | 52,1 | 75,7 | 68,2 | 86,3 | 29,0 | 74,6 | 73,4 | 89,9 | 68,1 | 94,4 | 50,1 | 90,9 | 56,6 | 8 | 4 | L12 |
| L13 | 68,9 | 73,8 | 66,3 | 48,6 | 66,3 | 39,4 | 63,8 | 43,2 | 80,5 | 71,3 | 46,2 | 26,7 | 68,2 | 42,7 | 76,1 | 52,2 | 76,7 | 55,1 | 79,7 | 30,0 | 70,5 | 28,0 | 6 | 2 | L13 |
| L14 | 69,4 | 56,4 | 52,4 | 46,8 | 56,3 | 43,5 | 64,6 | 43,6 | 82,3 | 60,5 | 49,7 | 38,9 | 63,8 | 32,0 | 55,7 | 58,5 | 57,0 | 32,1 | 61,7 | 21,8 | 55,7 | 19,0 | 6 | 0 | L14 |
| L15 | 54,6 | 63,5 | 58,0 | 48,5 | 54,4 | 63,2 | 51,7 | 81,0 | 70,1 | 76,3 | 64,7 | 67,7 | 59,8 | 37,4 | 66,7 | 60,0 | 20,2 | 60,0 | 53,8 | 41,9 | 55,9 | 47,2 | 4 | 3 | L15 |
| L16 | 54,2 | 86,9 | 86,0 | 58,9 | 95,1 | 89,6 | 67,5 | 81,1 | 97,2 | 38,5 | 64,0 | 80,9 | 92,2 | 58,6 | 73,4 | 80,9 | 84,0 | 91,5 | 94,5 | 62,9 | 95,1 | 66,0 | 8 | 5 | L16 |
| L17 | 76,1 | 68,3 | 82,2 | 59,4 | 77,7 | 61,3 | 78,9 | 65,2 | 91,5 | 77,2 | 38,9 | 46,0 | 73,7 | 47,5 | 66,6 | 62,1 | 84,8 | 67,3 | 92,3 | 46,2 | 89,4 | 41,5 | 4 | 2 | L17 |
| L18 | 67,5 | 72,6 | 81,7 | 57,9 | 68,7 | 66,4 | 61,7 | 74,7 | 87,4 | 86,9 | 46,3 | 75,4 | 62,4 | 64,4 | 68,7 | 84,5 | 66,5 | 49,2 | 81,7 | 56,3 | 82,5 | 54,0 | 5 | 1 | L18 |
| L19 | 35,3 | 55,2 | 51,7 | 60,5 | 55,7 | 54,2 | 90,0 | 69,7 | 92,0 | 72,1 | 59,1 | 56,4 | 74,3 | 64,0 | 66,6 | 49,0 | 48,9 | 72,1 | 75,5 | 63,8 | 79,4 | 52,8 | 5 | 2 | L19 |
| L20 | 60,7 | 50,0 | 25,2 | 27,0 | 15,8 | 62,9 | 39,1 | 49,1 | 63,7 | 26,0 | 65,1 | 40,6 | 30,0 | 34,6 | 51,3 | 67,9 | 34,2 | 16,4 | 28,1 | 24,3 | 47,6 | 26,2 | 1 | 0 | L20 |
| L21 | 40,7 | 46,2 | 33,2 | 29,5 | 26,0 | 13,0 | 33,7 | 46,8 | 61,5 | 76,6 | 45,1 | 39,2 | 40,1 | 65,9 | 46,5 | 22,5 | 60,6 | 42,1 | 60,6 | 55,6 | 49,1 | 44,2 | 1 | 2 | L21 |
| L22 | 30,8 | 45,1 | 3,7 | 41,3 | 12,0 | 33,3 | 43,6 | 91,9 | 76,1 | 60,0 | 64,1 | 81,3 | 11,0 | 67,6 | 21,3 | 85,6 | 8,1 | 74,8 | 10,4 | 56,6 | 9,8 | 61,7 | 0 | 3 | L22 |
| L23 | 43,5 | 83,4 | 38,0 | 58,2 | 44,9 | 82,6 | 65,6 | 96,9 | 85,6 | 92,9 | 70,4 | 68,8 | 44,1 | 78,2 | 59,6 | 72,5 | 22,5 | 87,9 | 16,7 | 60,9 | 25,5 | 59,1 | 1 | 2 | L23 |
| L24 | 22,6 | 30,9 | 91,6 | 45,7 | 39,3 | 21,9 | 23,1 | 61,2 | 100,0 | 91,6 | 8,0 | 0,0 | 0,0 | 0,0 | 62,3 | 61,2 | 52,2 | 77,0 | 77,9 | 100,0 | 54,8 | 100,0 | 1 | 2 | L24 |
| TOTAL | 11 | | 13 | | 19 | | 33 | | 28 | | 10 | | 7 | | 21 | | 19 | | 26 | | 20 | | | | |
| T_Season | 8 | 3 | 11 | 2 | 12 | 7 | 15 | 18 | 15 | 13 | 7 | 3 | 6 | 1 | 16 | 5 | 12 | 7 | 16 | 10 | 12 | 8 | | | |
| T_Ocean | 0 | 0 | 1 | 0 | 0 | 0 | 4 | 4 | 2 | 3 | 2 | 1 | 0 | 0 | 4 | 0 | 1 | 1 | 2 | 1 | 1 | 1 | | | |
| T_Land | 8 | 3 | 10 | 2 | 12 | 6 | 11 | 14 | 13 | 10 | 5 | 2 | 6 | 1 | 12 | 5 | 9 | 7 | 15 | 9 | 10 | 7 | | | |

**Fig. 2 | Temperature Variability Evaluation Matrices.** Temperature Variability Evaluation Matrix for different ocean (O1–O9) and land (L1–L24) regions for December, January, February (DJF) and June, July, August (JJA) months for the 11 single model initial-condition large ensembles (SMILES) for detrended (**a**) and non-detrended (**b**) temperature anomalies. Shading marks the fulfillment of Criteria 1, that the rank histogram of spatially averaged observations falls within the perfect-model rank range. Numbers mark the percentage of grid cells in the region that exhibit an unbiased representation of monthly surface air temperature (TAS) GISTEMPv4 observations over land and sea surface temperature (SST) ERSSTv5 observations over the oceans. An unbiased simulation at the grid cell level corresponds to observed monthly values exceeding the ensemble maxima or minima less than 8% of the months respectively, and occurring with the central 75th percentile ensemble bounds no more frequently than 80% of the months. A region and season are considered to be adequately simulated by a given model when it fulfils Criteria 1 (green and gray shading) and Criteria 2 (exhibiting at least 50% of the grid cells are unbiased). When both criteria are met the field is shaded in green, when only Criteria 1 is met the field is shaded in gray (see "Methods" for further details on the evaluation framework). The four bottom rows show the total number of regions that each model simulates adequately (TOTAL), as well as per season (T_Season) and across ocean (T_Ocean) and land (T_Land) regions, respectively. The last two columns show the number of models that adequately simulate a region per season. Regions for which less than 3 SMILEs offer adequate simulations are highlighted in red.

change the sign of the projected change in temperature variability, it can help to reduce the uncertainty in these changes, reducing the potential range of magnitudes of the change, particularly in regions that are poorly represented in climate models.

How this multi-model spread or uncertainty in the magnitude of the change (errorbars in Figs. 3 and 4) changes between the constrained and unconstrained ensembles is somewhat complex. The errorbars themselves show the multi-model ensemble spread, and are thus determined by the end members of the multi-model ensemble.

The interquartile range of the multi-model ensemble is shown in the wider bars. By design, the errorbars only decrease when the end members of the ensemble are removed. Therefore, only if our evaluation proves the more extreme model members to not perform adequately, and thus removes them from the constrained ensemble, does the uncertainty in the projections decrease. This becomes more likely to happen by chance with the removal of more models, but is not a predetermined nor expected outcome. Generally, constraining the projections to best-performing models yields a decrease in model

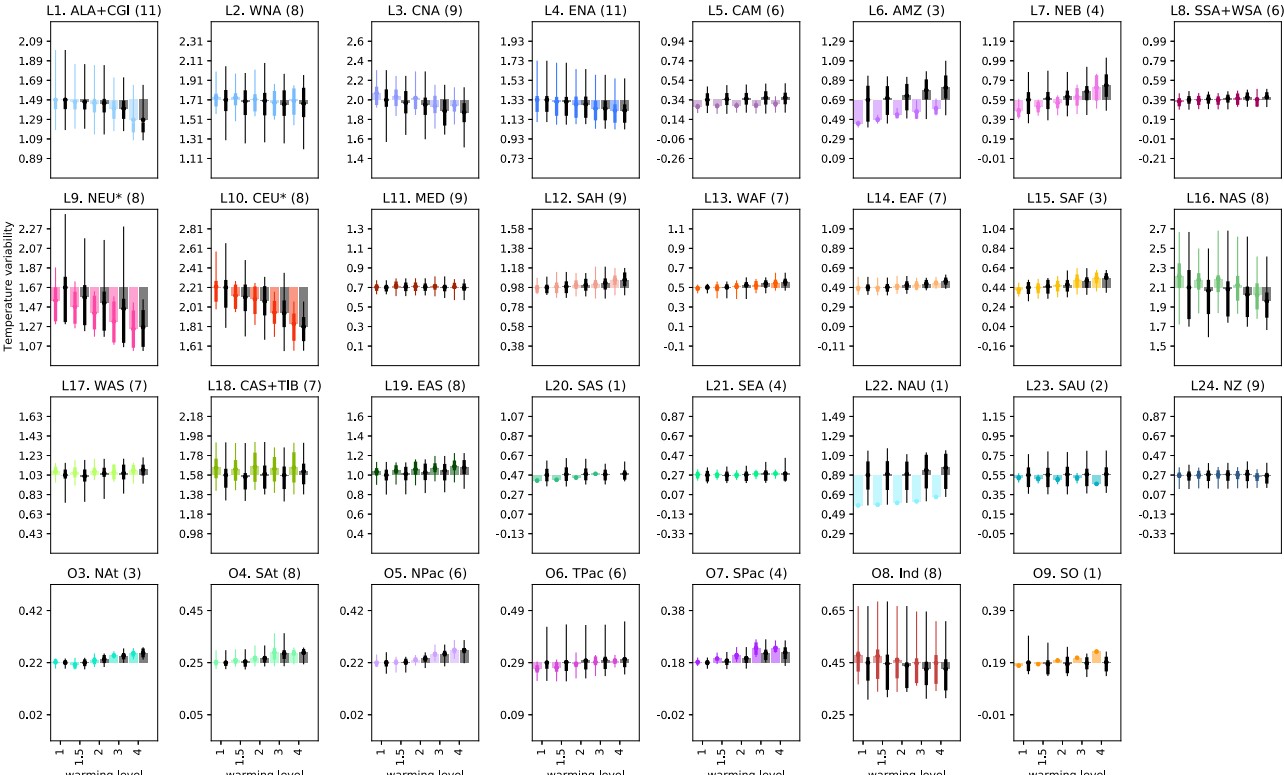

**Fig. 3 | Multi-ensemble mean December, January, February (DJF) temperature variability defined as the standard deviation over each region at each warming level for both the full ensemble (black) and the constrained ensemble (color).** Results are shown relative to the variability at 1 degree of warming in the full ensemble. Errorbars show the full model spread (i.e. the end members of each ensemble), with the fatter errorbars highlighting the 25th and 75th percentiles. The number of models that accurately represent observed variability are shown in the titles. Note we choose to exclude the polar oceans from these plots due to issues around the ice edges.

spread, thus an increase in model agreement and improved (as in less uncertain) projections. For regions where substantially fewer models are used in the constrained ensemble (6 or less), the model spread decreases substantially, as seen for example for the Amazon (L6) in both seasons or Southern Australia (L23) in DJF or the Tropical Pacific Ocean (O6) in JJA. However, this is not always the case, as seen in North Atlantic (O3) in DJF and Western North America and Central North America (L2 and L3) in JJA where the constrained ensemble consisting of 3 and 6 adequately performing models respectively has an almost identical spread as the unconstrained 11-model ensemble. We note that this exception is, albeit interesting, rare. Over several regions, the constrained projections indeed show an increase in model agreement, even for a large number of selected best-performing models. This can be seen for example in Northern Europe (L9) and Central North America (L3) in DJF with 8 and 9 models respectively included or in Northern Asia (L16) and Southern Australia (L23) in JJA both with 8 models included. These results indicate that performance constraining can indeed decrease the uncertainty in future variability projections, although this uncertainty is not eliminated completely, with decreased yet substantial remaining multi-model spreads in the constrained ensemble for most adequately modeled regions and uncertainties remaining large over key, poorly-modeled regions.

The magnitude of the temperature variability at 1 °C (Fig. 5; DJF & Fig. 6; JJA−left column panels) in both seasons is typically lower in the constrained compared to the full ensemble also at the grid-cell level, except for in the northern hemisphere extratropics in DJF. This highlights the improved representation of the present day variability, as the lower variability in the constrained ensemble over land is in better agreement with observations, except for again in the northern hemisphere extratropics in DJF (Fig. 7). We note that the variability over the ocean shows little improvement in the constrained ensemble, and tends to be overestimated in DJF and underestimated in JJA. While the variability itself is lower in the constrained ensemble, the change in variability from 1° to 3 °C of global warming increase (Fig. 5; DJF & Fig. 6; JJA−second column panels) tends to be larger in the constrained ensemble compared to the unconstrained. In particular, the variability in the constrained ensemble is much larger in both seasons over Northern South America and is somewhat larger over central Africa, Australia, and the extratropical Indian Ocean in JJA. The change in variability itself in both ensembles is typically positive in sign except for most of the northern hemisphere extratropical and higher latitude land masses and parts of the ocean in DJF in agreement with previous work[1,2,12,14–21].

Over three key areas that are typically poorly represented in many climate models, namely Australia & South-East Asia, South America, and Africa (Figs. 5 and 6), temperature variability is overestimated at the 1 °C warming level as compared to observations (Fig. 7). This overestimation, while present in both seasons, is largest in the local summer: for Australia and central South America in the austral summer (DJF), and larger in Northern Africa in JJA. This overestimation, while still present, is smaller in the constrained ensemble (Fig. 7). While the mean estimate of temperature variability is overestimated, the projected variability change from 1–3 °C is typically larger in the constrained ensemble over these regions. This overestimation of the variability itself at 1 °C and underestimation of the change from 1 °C to 3 °C manifests as larger variability in the full ensemble at 3 °C than the constrained ensemble in these regions (Fig. 7 left panels). This implies that the absolute value of the variability at 3 °C is overestimated in the full ensemble, but the change in variability may be underestimated, meaning that the interpretation of results could be biased in either

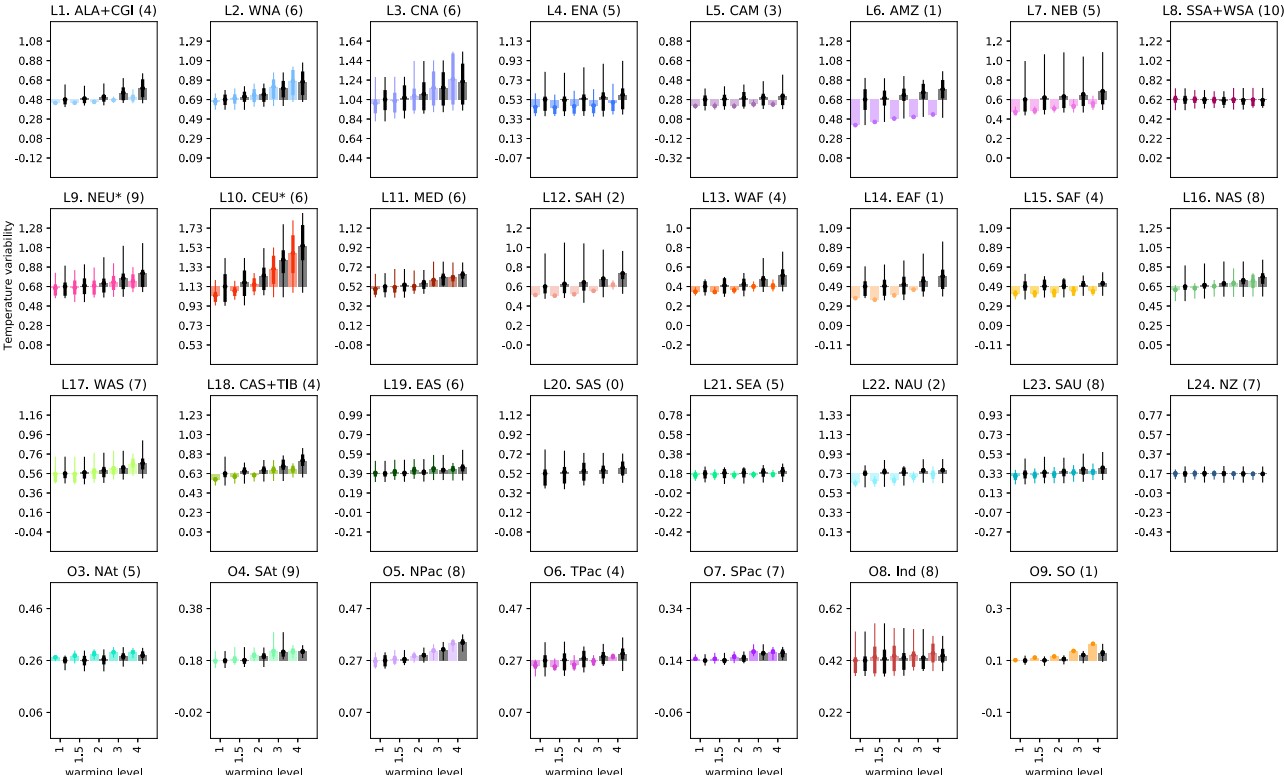

**Fig. 4 | Multi-ensemble mean June, July, August (JJA) temperature variability defined as the standard deviation over each region at each warming level for both the full ensemble (black) and the constrained ensemble (color).** Results are shown relative to the variability at 1 degree of warming in the full ensemble. Errorbars show the full model spread (i.e. the end members of each ensemble), with the fatter errorbars highlighting the 25th and 75th percentiles. The number of models that accurately represent observed variability are shown in the titles. Note we choose to exclude the polar oceans from these plots due to issues around the ice edges.

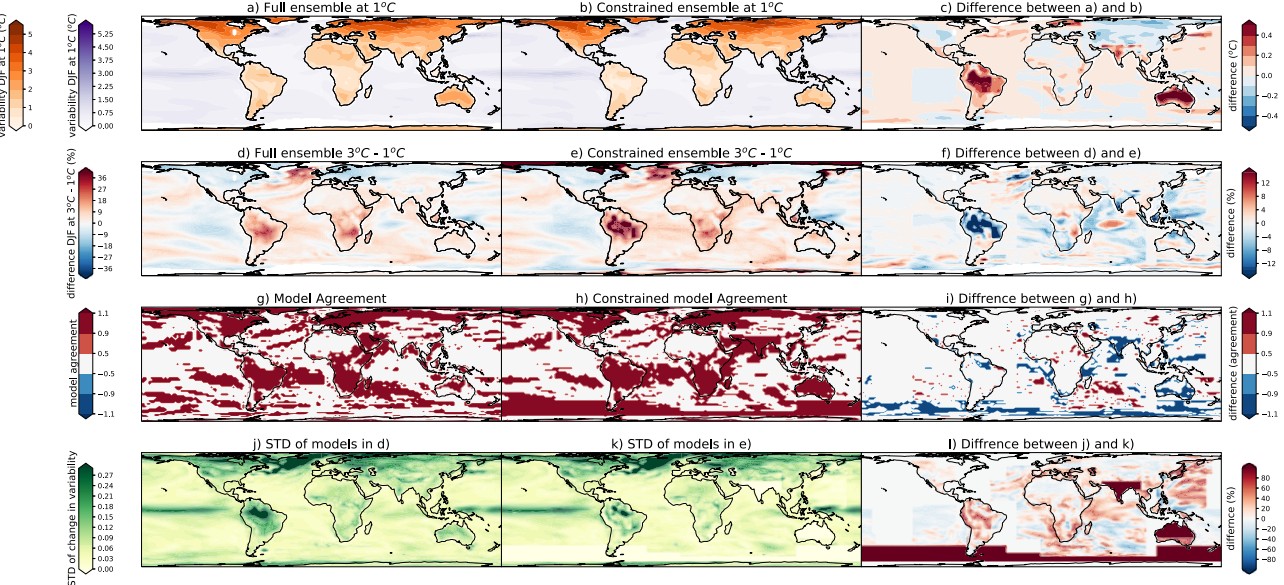

**Fig. 5 | The difference between the full and constrained ensembles in December, January, February (DJF). a** Temperature variability defined as the standard deviation over each region at 1 °C averaged across the full ensemble, **b** temperature variability at 1 °C averaged across the constrained ensemble, **c** difference between (**a**) and (**b**), **d** difference in temperature variability between 3 °C and 1 °C averaged across the full ensemble, **e** difference in temperature variability between 3 °C and 1 °C averaged across the constrained ensemble, **f** difference between (**d**) and (**e**), **g** model agreement on the sign of the change (red = agreement, white = disagreement) in the full ensemble, **h** model agreement on the sign of the change in the constrained ensemble, **i** difference between (**g**) and (**h**), **j** model agreement in the magnitude of the change (standard deviation across the full ensemble in the difference in temperature variability between 3 °C and 1 °C), **k** model agreement on the magnitude of the change in the constrained ensemble, **l** difference between (**j**) and (**k**) (shown as a percentage of the standard deviation (STD) of the full ensemble; **j**). Note in **a** and **b** two colorbars exist one over the ocean (purple) and one over the land (orange).

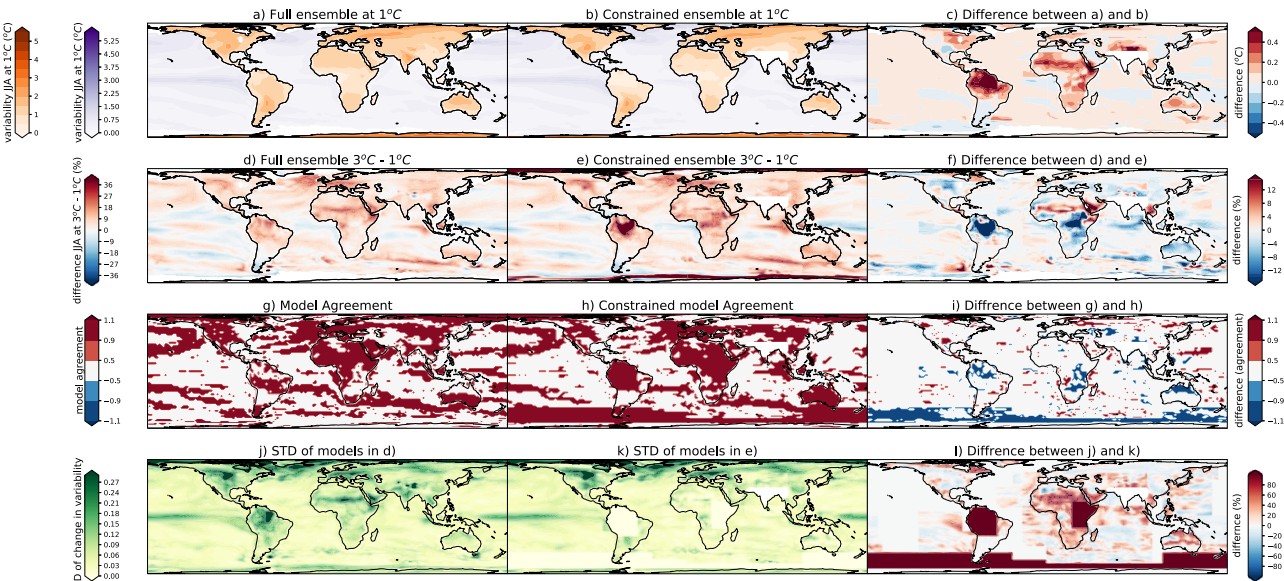

**Fig. 6 | The difference between the full and constrained ensembles in June, July, August (JJA). a** Temperature variability defined as the standard deviation over each region at 1 °C averaged across the full ensemble, **b** temperature variability at 1 °C averaged across the constrained ensemble, **c** difference between (**a**) and (**b**), **d** difference in temperature variability between 3 °C and 1° averaged across the full ensemble, **e** difference in temperature variability between 3 °C and 1° averaged across the constrained ensemble, **f** difference between (**d**) and (**e**), **g** model agreement on the sign of the change (red = agreement, white = disagreement) in the full ensemble, **h** model agreement on the sign of the change in the constrained ensemble, **i** difference between (**g**) and (**h**), **j** model agreement in the magnitude of the change (standard deviation across the full ensemble in the difference in temperature variability between 3 °C and 1 °C), **k** model agreement on the magnitude of the change in the constrained ensemble, **l** difference between (**j**) and (**k**) (shown as a percentage of the standard deviation (STD) of the full ensemble; **j**). Note in **a** and **b** two colorbars exist one over the ocean (purple) and one over the land (orange).

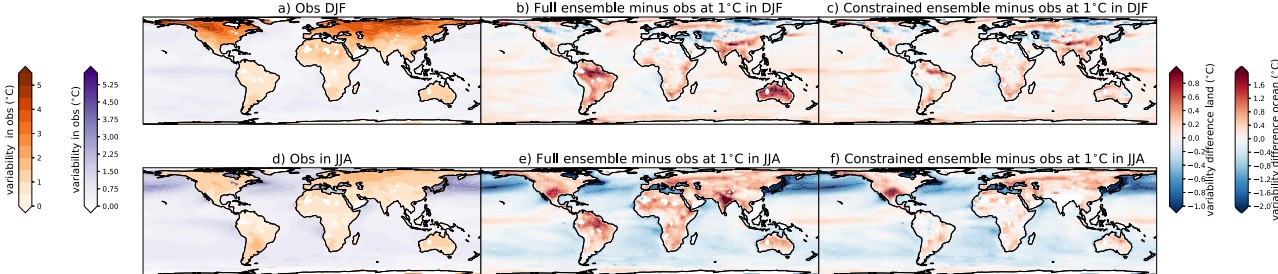

**Fig. 7 | Comparison of the full and constrained ensembles at 1 °C with observations.** Observed estimate of temperature variability defined as the standard deviation across the observed time series pooling all 3 months of data (**a**: December, January, February; DJF, **d**: June, July, August; JJA). Full ensemble estimate of temperature variability defined as the standard deviation at 1 °C of warming minus the observed estimate (**b**: DJF, **e**: JJA). Constrained ensemble estimate minus the observed estimate (**c**: DJF, **f**: JJA). Note the observations have been detrended using a second order polynomial fit. Observations are taken from ERSSTv5 over the ocean and GISS over the land as described in the "Methods". Note in **a** and **b** two colorbars exist one over the ocean (purple) and one over the land (orange). Differences over the ocean and land also have different colourscales shown by the colorbars on the right hand side of the plot.

direction depending on whether one computes the actual value of future variability or the value of the future change in variability. This is particularly true over South America in both seasons and central and South Africa and Australia in JJA, emphasizing that our projections of increased temperature variability and consequently temperature extremes may be underestimated in these poorly represented regions.

While we cannot assess whether the future projections are more realistic in the constrained ensemble compared to the full ensemble, as we do not have observations of the future, we can compare the projections with those from the most realistic model over these poorly modeled regions in the historical period. CESM2 is one of the two models that best represent Australia & South-East Asia (L21–23), South America (L6–8), and Africa (L12–16; the other being CESM-LE), with 15 "good performances" out of 22 across both regions and seasons. By using this model as a proxy for possible future observations, we can

compare the differences at 3 °C of warming between the full and constrained ensemble, the constrained ensemble (excluding CESM2), and CESM2 (Fig. 8; see Supplementary Information Figs. 3.1 and 3.2 for a spatially aggregated comparison across all regions). We find that, except for South America (where CESM2-LE fails to capture historical variability according to our evaluation for L6 and L7) and Africa in DJF the constrained estimate is closer to CESM2 (our "good model") than the full ensemble, tentatively suggesting that our future projections might be more realistic in the constrained ensemble compared to the full ensemble.

Finally, we assess model agreement in two ways. First, we consider agreement on the sign of the change. We find some patches of improved model agreement on the sign of the temperature variability change (Figs. 5 and 6, when panel i is blue) in India and Australia in DJF and small parts of South America, Africa and all of northern Australia in

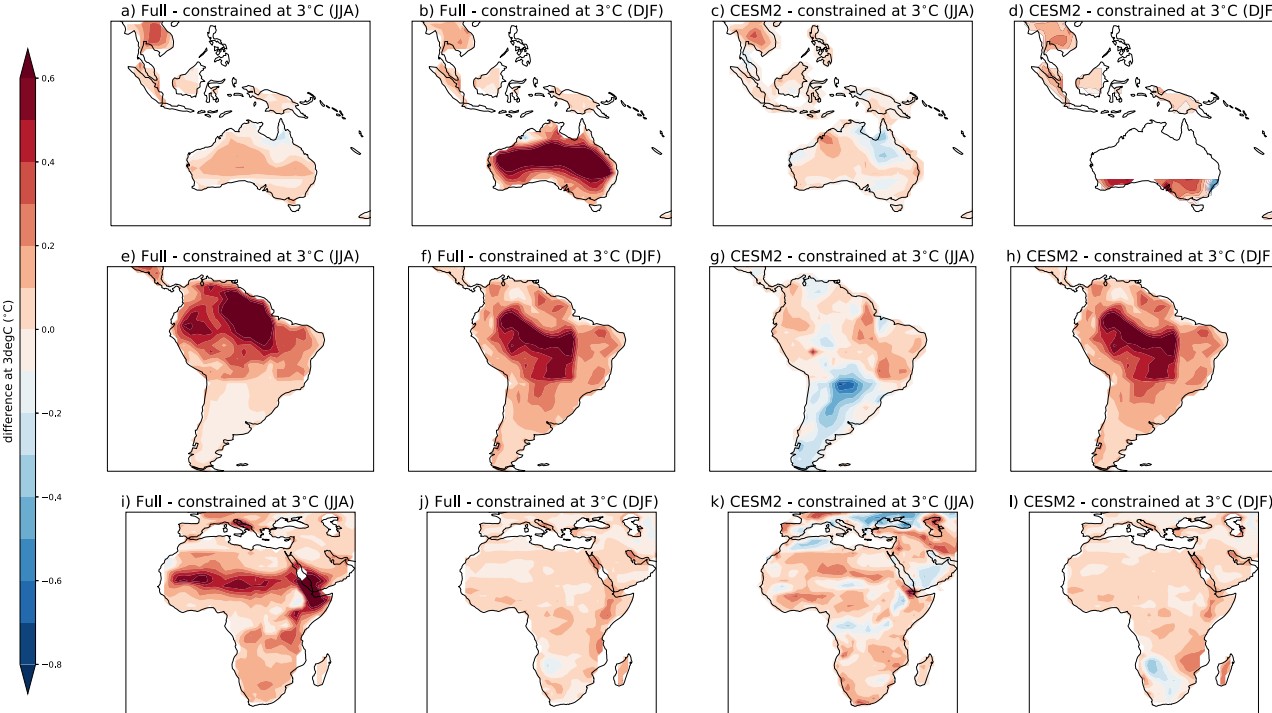

**Fig. 8 | Difference between the full and constrained ensembles over poorly represented land regions.** Full ensemble estimate of temperature variability defined as the standard deviation over each region minus the constrained estimate at the 3 °C warming level (left 2 columns; **a**, **b**, **e**, **f**, **i**, **j**). The right two columns (**c**, **d**, **g**, **h**, **k**, **l**) show the same thing but with the CESM2 estimate of variability minus the constrained ensemble (with CESM2 excluded). Shown for poorly represented land regions, Australia and South-East Asia (top row; **a**–**d**), South America (middle row; **e**–**h**), and Africa (bottom row; **i**–**l**) for both June, July, August (JJA; left) and December, January, February (DJF; right).

JJA. Next, we assess model agreement on the range of potential magnitude of the change. This is assessed by taking the standard deviation across the multi-model ensemble constituted by each model's mean change in temperature variability. Regions with high uncertainty in the magnitude of projected temperature variability change (Figs. 5 and 6j, k−darker green colors) in both seasons are the Tropical Pacific Ocean, Southern Ocean, Northern South America and the northern Hemisphere extratropical land surface. We find that the uncertainty in the magnitude of the change is similar or larger in the full ensemble compared to the constrained ensemble, except for in patches of the ocean. Regions that show substantially less uncertainty in the magnitude of the change in the constrained ensemble are South America, India, Australia & South-East Asia, the west Pacific Ocean and Southern Ocean in DJF and north South America, Africa, parts of Australia and the Southern Ocean in JJA. Typically, regions where model agreement on the sign of the change is substantially improved are regions where the constrained ensemble includes many fewer models than the full ensemble. Note, however, that this is not a predetermined result. We tested the change in the standard deviation (understood as the uncertainty in the magnitude of the change) in a constrained ensemble of either 4 or 6 randomly selected models by sampling 1000 times, and find the standard deviation decreased approximately 2/3 of the time and increased 1/3 of the time.

In summary, our results highlight that constraining by historical model performance in the past does not necessarily improve model agreement in the future. However, while the agreement on the sign of the change is not drastically improved by the constraint, the spread of projections and hence the uncertainty in the variability change can be limited using this method. This shows that selecting models that best simulate historical variability can still be useful, albeit not a silver bullet that can be used to rectify all model differences and uncertainty in future temperature variability change projections.

## Discussion

In this study we produce projections of future temperature variability based on simulations from the available SMILEs, and constrain them to those SMILEs that best capture historical temperature variability in observations over different regions and seasons. This work expands on previous rank-frequency evaluation frameworks[33] by including two formally designed evaluation criteria. These criteria determine whether models adequately capture historical observations, considering the limited sampling of internal variability allowed by the relatively short observational record, by implementing a perfect-model evaluation set up. This novel and rigorous variability evaluation enables end-users of these SMILEs to select the best-fitting model for their region and season of interest. Furthermore, it provides a frame of multi-model reference, as many large ensemble studies are currently still based on one single model or on a very limited number of models. We urge caution in using projections from a single model, as we find that some widely used models show a rather poor performance in capturing observed temperature variability and observed forced changes, and should therefore be used with care.

Our evaluation reveals CESM-LE, CESM2-LE and GFDL-SPEAR-MED as the Top 3 best performing models in capturing isolated temperature variability (i.e., using detrended temperature data as base for the evaluation). The Top 3 best performing models capturing both temperature variability and forced changes in the historical record are CESM-LE, CESM2-LE, and MPI-GE5. We also find that temperature variability is systematically not well represented by all or most models across both seasons over large areas, e.g., the Arctic (O1), Antarctic (O2), and Southern (O9) Oceans, or the Indian Peninsula (L20) and Northern Australia (L22) for both seasons, as well as the Amazon basin (L6), Northern (L12) and Eastern Africa (L14), and Southern Australia (L23) specifically in the local summer season. In contrast, some areas are systematically well captured by most models, especially in the local

winter season, e.g., North America (L1–L4), Southern South America (L8), Europe and Northern Africa (L9–L11) or North and Central Asia (L16–L19).

Our results highlight that models typically overestimate temperature variability in the current climate. In some regions, particularly in South America in both seasons, and parts of Africa, Australia in JJA, they may, however, underestimate the projected temperature variability change. While the uncertainty in the sign of the change is generally unchanged by constraining the ensemble, the performance-based constraint presented here can help reduce the ensemble spread and reduce the range of possible magnitudes of the projected change, providing a performance-constrained estimate of both the projected change and the model uncertainty around it. This improvement in model agreement in both the sign and magnitude of the change typically (but not only) occurs in regions where the models underperform, and hence the constrained ensemble contains many less members than the full ensemble (e.g. Australia, Africa, and South America). In these regions we find that temperature variability itself is typically overestimated in present day climate; while the constrained ensemble provides a larger projected change in variability (Figs. 5 and 6). This means that variability projections that ignore model performance in capturing current variability yield underestimated variability increases in these regions, and thus may underestimate increases in warm temperature extremes compared to the performance-constrained ensemble. Furthermore, the overestimation of current climate variability combined with underestimation of its increase suggests that our models may overestimate present day potential extremes This has implications for planning for extreme events. These results are vital for producing robust future projections of extremes and to inform adequate adaptation strategies.

We highlight that models perform more adequately in the local winter season. This is cause for concern, as for many regions the local summer season that is less well simulated reflects the period when temperature extremes and temperature variability changes are likely to be the most impactful. The set of regions and seasons where temperature variability is captured incorrectly by all or most models according to our evaluation metrics paints an unsettling picture. These regions, including the Amazon basin and Indian Peninsula in both seasons, Central America and East Africa in JJA, and the Maritime Continent and Northern Australia in DJF, are some of the most populous, vulnerable and environmentally diverse regions on Earth, and the lack of adequate simulations of current climate temperature variability estimates endangers not only the reliability of our projections of future variability and its change, but also our base knowledge on the potential climatic, socioeconomic and ecological impacts that climate change may bring over these areas.

Our work highlights that, while constraining by historical model performance in the past can reduce the spread of temperature variability change projections and hence reduce model uncertainties, these uncertainties are not completely erased. Even when successfully identifying several models that perform adequately in capturing today's climate in certain regions, model disagreement in temperature variability projections remains, in many cases, large. Thus, model differences across temperature variability change under future climatic conditions remain, over large areas, unreconciled. We urge the community to continue working on constructing and improving upon climate models that better capture real-world processes, while producing independent projections of future climates that sufficiently sample inter-model uncertainties that cannot yet be reduced. Our work highlights that the climate science community cannot yet afford to move away from having several independent climate models to sufficiently sample model uncertainty. As modeling centers across the world are pooling resources and developing fewer, less independent models, we risk losing sight or the range of possible futures that we need to be prepared for. Our attempt to constrain projections of future

temperature variability change based on historical performance is an example of a change in the climate system that cannot yet be foreseen with the best available knowledge.

Therefore, we can summarize the main conclusions and implications from this study as follows:

- We provide a comprehensive evaluation of the state-of-the-art SMILEs ability to represent the historical summer and winter temperature variability in observations. We identify CESM-LE and CESM2-LE as the SMILEs that provide the best representation of isolated temperature variability as well as of both temperature variability and forced change, with GFDL-SPEAR-MED and MPI-GE5 as close third in each category respectively. This multi-model evaluation across all available CMIP5 and CMIP6 SMILEs provides a basis for model selection for the assessment of temperature variability and extremes, which cannot be correctly simulated if the underlying variability is incorrect.

- Our evaluation also shows that some regions are systematically not well represented such as the Southern Ocean (O9), the Indian Peninsula (L20), and Northern Australia (L22), as well as the Amazon basin (L6), Northern (L12) and Eastern Africa (L14), and Southern Australia (L23), all specifically in the local summer season. We also find that certain areas of the world are systematically well captured by most models, especially in the local winter season, namely North America (L1–L4), Southern South America (L8), Europe and Northern Africa (L9–L11) and North and Central Asia (L16–L19). Finally, we conclude that local winter seasons are generally simulated better by most models more often than local summer seasons.

- Similar to the IPCC, we find low model agreement on the sign of the change over large parts of the Earth's surface, and large model disagreement on the magnitude of the projected change under future warming, particularly over the land surface and the tropical Pacific Ocean. This implies that we cannot afford to move away from multi-model ensembles that sufficiently capture the uncertainty in our future projections.

- The constrained ensemble decreases model spread over some regions and gives a lower range of projected futures, providing a smaller range of potential future projections and greater model agreement, particularly in South America, India, Australia & South-East Asia, the west Pacific Ocean and Southern Ocean in DJF and north South America, Africa, parts of Australia and the Southern Ocean in JJA. However, the constraint does not substantially increase model agreement on the sign of the projected change.

## Methods
### Climate model simulations and observational data
We include 11 SMILEs from a broad range of climate models across different CMIP generations: ACCESS-ESM1.5[41], CanESM2[42], CanESM5[43], CESM-LE[44], CESM2-LE[45], CSIRO-MK3.6[46], GFDL-ESM2M[47], GFDL-SPEAR-MED[48], MIROC6[49], MPI-GE5[50] and MPI-GE6[51].

Each SMILE consists of many simulations for each climate model that differ only in their initial state, and evolve under the same specific forcing conditions. However, the SMILEs differ in the number of simulations included (from 30 up to 100 members), in their rate of warming under increasing anthropogenic emissions (Equilibrium Climate Sensitivity, ECS, values of 2.4 K to more than 5 K), in the initialization method (from micro atmospheric perturbations to different initial states sampled from the control simulation), in the generation of forcing scenarios used (CMIP5 to CMIP6), and in the forcing scenario (historical simulations are extended with a high emissions scenario such as RCP8.5, SSP370 or SSP585). More details can be found in Table 1, and in previous studies[33,50,52,53] and references therein.

Observed surface air temperature data from the GISSTEMPv4[39] dataset for the period of 1880-2024 and sea surface temperature data

**Table 1 | Details of single model initial-condition large ensembles (SMILE) experiments included**

| SMILE | Members | Years | Gen. | Forcing | ECS | Reference |
|---|---|---|---|---|---|---|
| ACCESS* | 40 | 1850–2100 | CMIP6 | Hist + SSP585 | 3.9 K | 41 |
| CanESM2 | 50 | 1950–2100 | CMIP5 | Hist + RCP8.5 | 3.7 K | 42 |
| CanESM5* | 50 | 1850–2100 | CMIP6 | Hist + SSP585 | 5.7 K | 43 |
| CESM-LE | 40 | 1920–2100 | CMIP5 | Hist + RCP8.5 | 4.1 K | 44 |
| CESM2-LE* | 100 | 1850–2100 | CMIP6 | Hist + SSP370 | 5.1 K | 45 |
| CSIROMK3.6 | 30 | 1850–2100 | CMIP5 | Hist + RCP8.5 | 4.1 K | 46 |
| GFDL-ESM2M | 30 | 1861–2100 | CMIP5 | Hist + RCP8.5 | 2.4 K | 47 |
| GFDL-SPEAR-MED* | 30 | 1921–2100 | CMIP6 | Hist + SSP585 | 1.8 K | 48 |
| MIROC6* | 50 | 1850–2100 | CMIP6 | Hist + SSP585 | 2.6 K | 49 |
| MPI-GE5 | 100 | 1850–2099 | CMIP5 | Hist + RCP8.5 | 2.8 K | 50 |
| MPI-GE6* | 50 | 1850–2100 | CMIP6 | Hist + SSP585 | 2.8 K | 51 |

Experiment name, number of members, simulated years used, forcing generation, forcing scenarios, and Equilibrium Climate Sensitivity (ECS) of SMILE experiments included in our study. All experiments include historical forcing (Hist) until 2005 for CMIP5 or until 2014 for CMIP6. CMIP6 generation SMILEs are marked by a star. ECS refers to the equilibrium temperature response to the doubling of carbon dioxide[43,50,58,59]. Note that CESM2-LE consists of two 50-member sets with slightly varying biomass burning emissions forcing fields, that we treat as a single 100 member ensemble[45].

from ERSSSTv5[54] from 1854–2024 are used for the evaluation and comparison to the SMILE simulations. We define surface air temperatures (TAS) as the near-surface 2 m air temperature anomaly over land grid cells, and sea surface temperatures (SST) as the surface temperature over ocean grid cells. For the purpose of comparing model simulations and observations, all simulated data are regridded to the coarser resolution of GISSTEMPv4 (180 × 90) and ERSSTv4 (180 × 89) observations, and subsampled to grid boxes where observations are available. Both observed and simulated data are compared as anomalies calculated with respect to the climatological baseline defined by the period 1961–1990 in each ensemble member and observations.

We define 24 land regions and 9 ocean regions (Fig. 1k, l) roughly following the IPCC SREX region definition[55]. Some regions have been merged or sightly reshaped to avoid having regions consisting of too few land or ocean grid cells, respectively.

As projections of temperature variability depend on both the greenhouse gas emissions and the climate sensitivity of the model[22]; we base our multi-model comparison on warming levels rather than selecting specific time periods. This allows for a comparison that is independent of how fast a model warms. Warming levels are shown in Supplementary Information Table 1, and defined relative to a baseline period of 1850–1899 in each SMILE. Each warming level is computed as the year when the global mean surface temperature (annual) from the SMILE ensemble mean crosses an individual warming level.

**Rank-frequency variability evaluation framework**

The rank-frequency evaluation framework applied in this study determines how well models capture the internal variability in observations based on a simple principle: whether the range of climate states, in this case defined by winter and summer monthly mean temperature anomalies, as simulated by a climate model agrees well with the range that observations cover in the historical period. This framework allows us to assess a model's performance in simulating the variability in observations without the need to parametrize or make any assumptions regarding the shapes of the observed or simulated distributions. The basis of this evaluation framework, which resembles probabilistic forecast verification techniques in the climate prediction literature[34–36], is demonstrated for annual mean temperatures and 10 SMILEs in ref. 33. It was first developed to evaluate European summer temperature and precipitation in MPI-GE5 in ref. 56; Supporting Information Figs. S1–S3; and further expanded globally for MPI-GE5 in ref. 50 and in ref. 11 for annual mean temperatures and summer maximum temperatures, respectively, as well as in ref. 57 for

temperatures over North America in six SMILEs. For further details on the existing framework and its theoretical justification and interpretation based on idealized and specific examples see ref. 33.

Here, we expand on this existing evaluation framework by incorporating two formally defined evaluation criteria, which again assess model performance based on whether observations occur uniformly across all ensemble ranks of an SMILE (i.e., the position an observation takes among the sorted ensemble members for a given time step). The first criteria, based on spatially averaged metrics, determines whether observations occur across all rank windows with uniform frequency. Traditionally, this has been assessed in the ensemble forecasting literature by evaluating rank histogram *flatness*[35,37]. However, the relatively short length of the observational record means that the influence of internal variability may not be robustly sampled on these timescales. This potentially insufficient variability sampling, combined with the fact that we are comparing observations against uninitialised, free-running model simulations means that, with the sample size available, rank histograms may not be perfectly flat, even for perfectly performing models.

To overcome this, we apply a perfect-model rank range test. This range is constructed by treating each ensemble member in each SMILE as if it were observations, and calculating the resulting spread of model rank histograms. This approach provides a relatively wide distribution of rank histograms that reflect perfect-model behavior, yet may indeed not be perfectly flat. These deviations from flatness in a perfect-model set up can occur either systematically because the simulated temperature distribution is skewed or non-normal, or because the record length is too short to sufficiently sample rank variability, or both. Similarly, the observational rank histogram may also appear to be non flat for the same reasons, without necessarily implying incorrect model performance. Since in a perfect-model setup each member captures its model behavior perfectly, this rank range gives us a baseline of possible deviations that could occur due to insufficient internal variability sampling or other factors, even for models that capture observations adequately.

Therefore, we determine adequate model performance for spatially averaged metrics when the observations rank histogram is within the perfect-model rank histogram range (Criteria 1). In this test, we allow for a maximum deviation of 10% of the rank frequency value beyond the perfect-model range bounds (so for an upper perfect-model rank frequency bound of 3% for any particular rank, the maximum allowed rank frequency for observations would be 3.3%). To minimize the effect of varying ensemble sizes and potentially spurious effects resulting from the insufficient rank variability sampling due to

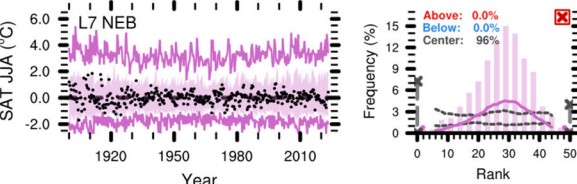
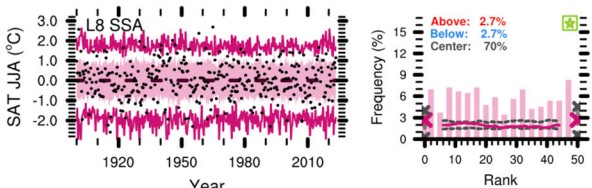

**Fig. 9 | Example of time series and rank frequency histograms for MIROC6 against GISTEMPv4 observations.** Time series and rank frequency histograms of spatially aggregated June, July, August (JJA) temperature anomalies (TAS) for land regions L7 (left panel) and L8 (right panel). Time series show the ensemble maximum and minimum (colored lines) and central 75th percentile ensemble spread (shading) against observations (black dots). Rank histograms show the frequency of each place that observations would take in a list of ensemble members ordered by ascending temperature anomaly values. Rank 0 indicates observations are below the minimum ensemble value, and rank $n$, with $n$ the number of ensemble members, indicates that observations exceed the maximum ensemble value for that particular month. For a model that perfectly represents observations over an infinitely-long observational record, all ranks should occur with uniform frequency and its histogram should be roughly flat. To illustrate how internal variability may affect rank frequencies given the non-infinite record length considered, we include a perfect-model rank range (gray), which shows the range of rank histograms that each ensemble member would yield if it were observations. Histogram bars

illustrate the individual rank frequency for observations, lines illustrate the running mean rank frequency over a centered $n/5$-bin window for observations (solid colored lines), and for the perfect model rank range (gray dashed lines). Crosses represent the frequency of minimum (0) and maximum ($n$) ranks for observations (colors), and for perfect model range (gray). If the rank frequency exhibited by observations (colors) is within this perfect-model range (gray) for all rank windows, with a maximum allowed deviation of 10%, the model fulfils Criteria 1 for this region and season, showing an adequate model performance for spatially aggregated metrics, and is highlighted by a green star at the top right. If it does not fulfil this criteria is it noted by a red cross at the top right. Percentages at the top left of the histograms show the frequency of monthly anomalies occurring above (red) or below (blue) ensemble limits, or clustering within the central 75th percentile range (gray), analogous to the thresholds for the grid-cell evaluation in Criteria 2. The full selection of time series and rank histogram figures for all regions, seasons and models can be found in the Supplementary Information.

the finite sample size, this test is performed over windows of aggregated ranks. Therefore rank frequencies are assessed as the running mean across individual rank frequencies over an $n/5$ rank window, with $n$ being the ensemble size (for $n = 50$ members, 10-rank running average) for ranks 1 to $n − 1$, and as the actual rank frequency for ranks 0 and $n$ (e.g., Fig. 9).

This first evaluation criteria based on spatially aggregated metrics is complemented by a second grid-cell level evaluation criteria, to account for biases at the grid-cell level that may be smoothed or compensated in this spatial average. This second test follows similar uniform rank frequency principles, but is simplified to allow for a computationally efficient grid-cell level performance assessment. Instead of assessing the full spectrum of ranks against a perfect-model range, we assess only the top, bottom and central rank sections against fixed frequencies assumed to be a reflection of an adequate rank distribution. Therefore, this test assesses whether observed anomalies at the grid-cell level cluster around the center bounds of the simulated ensemble spread or fall outside of its limits with too high frequencies.

We determine when these grid-cell level variability biases are present (Criteria 2), according to the following principles: (a) A variability overestimation bias is detected when observed values cluster within the central 75th percentile ensemble bounds (12.5th to 87.5th percentiles) more frequently than 80% of the time steps. This bias implies that the model simulations are systematically more extreme than observed values and the width of the distribution is overestimated by the model, and would result in a mound shaped rank histogram. (b) A variability underestimation bias is detected when observed values exceed the ensemble minima or maxima, also understood as observations taking minimum or maximum ranks respectively, more than 8% of the months. This bias implies that the model fails to simulate extreme enough events with adequate frequency, which means that observations exhibit a systematically wider distribution than the model, and would result in a concave shaped rank histogram. To account for non-symmetrical behavior over the warm and cold tails of the temperature distributions, we account for this bias in variability underestimation when observations fall too often below the ensemble minima or above the ensemble maxima separately. Lastly, when neither bias (a) nor (b) are present, we determine a grid-cell to be unbiased. Therefore for the second criteria, we consider a region and season to be adequately simulated by a given model at the

grid-cell level when at least 50% of the grid cells in the region are unbiased (Criteria 2).

Note that the variability underestimation threshold of 8% is chosen to be rather conservative to account for the different ensemble sizes and relatively short observational record considered in this evaluation. For a 50-member perfect-model ensemble assessed over an infinitely long observational record, observations would exceed the ensemble maxima, which marks on average a 1-in-50-years event, 2% of the time, or twice a century. Due to internal variability, this frequency may fluctuate in any given century, as seen in the minimum and maximum rank frequencies for perfect model ranges (gray crosses in the rank frequency histograms in the Supplementary Information evaluation or in Fig. 9) oscillating between 0 to over 10% for 50-member ensembles. Additionally, these perfect model ranges in the minimum and maximum rank frequencies also fluctuate for regions that exhibit non-normal or skewed behavior in their temperature distributions. This means that in some particular cases this variability underestimation threshold frequency may be larger than the theoretical expectations, even in a perfect model set up. For this reason, and in the interest of simplicity for our evaluation framework, we choose a fixed frequency of 8% in ensemble limit exceedance as a variability underestimation bias threshold. For a more in-depth assessment of these biases we recommend to account for the effect of ensemble size and assign a varying threshold for this criteria for each SMILE assessed, ideally depending on the ensemble size and following perfect-model behavior as guideline. Similarly, the variability overestimation threshold frequency determining how often observations may cluster within the central 75th percentile of the ensemble spread is set at 80%, and not at 75%, to account for this effect of internal variability.

Therefore, the two evaluation criteria used in our framework are as follows:

- Evaluation Criteria 1. Perfect-model rank range regional level performance: the observations rank histogram for regionally averaged temperatures must lie within the perfect-model rank range across all ranks (with a maximum 10% deviation).
- Evaluation Criteria 2. Threshold-based grid-cell level performance: at least 50% of the region's grid-cells must be unbiased, meaning, at the grid-cell level, (a) observations do not cluster excessively within the central percentiles of the ensemble (i.e., observations do not occur with more than 80% frequency within the central 75th percentile ensemble bounds, indicative of

variability overestimation bias) and (b) fall outside the ensemble range too frequently (i.e., observations do not occur with more than 8% frequency outside of the ensemble spread, indicative of variability underestimation bias).

Only when both criteria are fulfilled, we consider the model in question offers an adequate performance for the region and season considered, and is selected to be part of the constrained ensemble for the particular region and season.

These evaluation results are summarized in the temperature variability Evaluation Matrix for different ocean (O1–O9) and land regions (L1–L24) for DJF and JJA months for the 11 SMILES for detrended data used for our assessment constraint (Fig. 2a) and additionally for non-detrended data (Fig. 2b). Numbers mark the percentage of grid cells in the region that exhibit an adequate, unbiased representation of observations. The color shading marks cases that fulfill Criteria 1, with the observations rank histogram within the perfect-model range. For the regions and seasons that also fulfill Criteria 2 and are deemed to be adequately simulated by a given model at the grid-cell (at least 50% of the grid cells are non-biased), the fields are highlighted in green and considered adequately captured. When only Criteria 1 is fulfilled, meaning the rank of the spatially aggregated metric fits the perfect model range but 50% or less of the grid cells are unbiased, the fields are highlighted in gray and considered inadequate.

We also show the full results of this evaluation, including evaluation time series and rank histograms and results for Criteria 1 and the grid-cell level assessment maps for Criteria 2 for all 11 SMILEs separately, for the DJF and JJA seasons and for both detrended and non-detrended temperatures against GISTEMPv4 and ERSSTv5 observations, respectively for land and ocean areas, in the Supplementary Information.

For the purposes of this evaluation, we use monthly mean temperature anomalies relative to the period 1961–1990, and model output data are regridded to match the different observational grids for land- and ocean-based evaluations. All evaluations are performed over the period for which each observational record is available starting in 1900 until 2024, and restricted to the period when simulations are available for the models that span shorter periods than observations. To avoid spurious effects of in-homogeneous variability behavior over the Amazon region in some models over the first years of their simulations, Amazon land region (L6) evaluations start only from 1920 onward. To isolate variability biases from potential biases in the forced warming rate, we base our main performance assessment and constraint on detrended temperatures, and provide the analogous assessment on non-detrended temperatures for comparison. Detrending is done for model data by subtracting from each model member the model's ensemble mean, and by subtracting a least squares quadratic trend from observations, both at the grid-cell level. Note that, in contrast to subtracting the ensemble mean, subtracting a quadratic trend may not ensure a perfect removal of the forced signal from observations, and remaining forced effects in observations may contribute to some of the discrepancies found in our evaluation. Lastly, to allow a more finely resolved temporal analysis and to increase sample size we perform this rank-frequency evaluation on monthly mean temperature anomalies instead of seasonal averages.

### Variability estimates

Temperature variability is computed for each season at each of 5 warming levels as the standard deviation of temperature pooled from each of the 3-months in the season for 11-years centered on the year defined for each warming level. All other calculations use the methodology of ref. 23 to take multi-ensemble means and calculate changes in temperature variability. Calculations for the full ensemble use all data from all SMILEs, while calculations for the constrained ensemble

use data from the SMILEs deemed as adequate in the variability evaluation framework in Fig. 2a for each specific region and season. The errorbars in Figs. 3 and 4 are defined as the minimum and maximum values across the ensemble. Model agreement in the third column of Figs. 5 and 6 is defined as when greater than 50% (light blue and red) and 80% (dark blue and red) of the models in the ensemble agree on the sign of the change. Uncertainty in the magnitude of the change in the fourth column of Figs. 5 and 6 is shown as the standard deviation across each model's individual estimate in the ensemble (i.e. for the full ensemble across 11 estimates, one from each model). Results in Figs. 3–8 are shown as the multi-ensemble mean (MEM) of the individual SMILE results. Results are typically shown for the MEM of the full and the MEM of the constrained ensemble separately, with the difference also plotted. We note that Figs. 3–6 exclude regions Arctic (O1) and Antarctic (O2) as no models perform adequately here, however we are concerned sea-ice may influence our results in this region when calculating variability.

## Data availability
The SMILE data used in this study can be found on a common grid in the Multi Model large ensemble archive version 2 [MMLEAv2[53]] and downloaded from the NSF NCAR Geoscience Data Exchange https://www.cesm.ucar.edu/community-projects/mmlea/v2. GISSTEMPv4 can be downloaded from NASA-GISS at https://data.giss.nasa.gov/gistemp[39] and ERSSTv5 can be downloaded from the NOAA Physical Sciences Website https://psl.noaa.gov/data/gridded/data.noaa.ersst.v5.html[54].

## Code availability
All code used in this research can be found at https://doi.org/10.5281/zenodo.17058694. The analysis and figures in this article have been created using Climate Data Operator (CDO) software, Python and NCAR Command Language (NCL, NCAR 2019; Version 6.6.2).

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

## Acknowledgements
N.M. was supported by the Australian Research Council Discovery Early Career Researcher Award DE230100315. L.S.G. received funding from the European Union's Horizon Europe Framework Programme under the Marie Skłodowska-Curie grant agreement No. 101064940. We thank the Deutsches Klimarechenzentrum (DKRZ) for providing the necessary computational resources to carry out this work. We thank Sebastian Milinski for computing the warming levels used in this project and both Sebastian Milinski and Jochem Marotzke for their intellectual contributions to the development of this project in its early stages. We additionally thank Thomas Frölicher and Dirk Olonscheck for providing data and access for GFDL-ESM2M and early access to MPI-GE-CMIP6, respectively. We acknowledge the US CLIVAR Working Group on Large Ensembles for their provision of the Multi-Model Large Ensemble data and we the World Climate Research Programme, which, through its Working Group on Coupled Modelling, coordinated and promoted CMIP6. We thank the climate modelling groups for producing and making available their model output, the Earth System Grid Federation (ESGF) for archiving the data and providing access, and the multiple funding agencies who support CMIP6 and ESGF.

## Author contributions
N.M. devised the project idea, compiled the SMILE data, and computed the variability calculations. L.S.G. devised the evaluation methodology, and completed all evaluation calculations. N.M. and L.S.G. co-wrote and revised the manuscript.

## Funding

## Competing interests
The authors declare no competing interests.
