## [Transparent Peer Review file · Nature Communications]

Temperature variability projections remain uncertain after constraining them to best performing Large Ensembles of Individual Climate Models

Corresponding Author: Dr Laura Suarez-Gutierrez

Version 0:

Reviewer comments:

Reviewer #1

(Remarks to the Author)

Summary and main comments

The authors evaluated historical simulations of seasonal (DJF and JJA) temperature variability in 11 large ensembles compared to observations using rank-frequency analysis (known as rank histograms in the ensemble forecasting literature). Models whose historical performance was judged adequate (by region and season) were included in a performance-constrained ensemble. Projections based on the full and performance-constrained ensemble were compared.

1. The introduction provides few (one?) references to application of performance-based constraints/subsetting to projections. There is an extensive literature on this topic, and I think it would be helpful to connect the current work to it. The other topics (importance of temperature variability and SMILES) are well introduced. A arbitrarily chosen example is

Brunner, L., Pendergrass, A. G., Lehner, F., Merrifield, A. L., Lorenz, R., and Knutti, R.: Reduced global warming from CMIP6 projections when weighting models by performance and independence, *Earth Syst. Dynam.*, 11, 995–1012, <https://doi.org/10.5194/esd-11-995-2020>, 2020.

This would also provide an opportunity to give the reader a clear idea of what has been done before in this direction and how this work improves upon previous efforts. It may be the case that model weighting/subsetting has not been applied to questions of variability changes, and if so, pointing this out would highlight the value of the work. Or perhaps other approaches suffer from limitations that are addressed here. At present, I found the limitations of past work/goal of present work statement not very clear (line 43): "However, it remains unclear whether climate models disagree in the direction and intensity of temperature variability changes because they do not adequately simulate temperature variability over certain regions, or because of other, more systematic model differences in future temperature variability behaviour. In this study, we address this question by constraining projections of temperature variability in a multi-model super ensemble of single model initial-condition large ensembles (SMILEs) to models that adequately represent observed regional variability under current climate conditions."

2. Rank-frequency analysis or more commonly rank histograms are a standard metric for assessing the reliability of ensemble forecasts. Classical references are

Anderson, J. L., 1996: A method for producing and evaluating probabilistic forecasts from ensemble model integrations. *J. Climate*, 9, 1518–1530.

Hamill, T. M., and S. J. Colucci, 1997: Verification of Eta–RSM Short-Range Ensemble Forecasts. *Mon. Wea. Rev.*, 125, 1312–1327, [https://doi.org/10.1175/1520-0493\(1997\)125<1312:VOERSR>2.0.CO;2](https://doi.org/10.1175/1520-0493(1997)125<1312:VOERSR>2.0.CO;2).

Talagrand, O., R. Vautard, and B. Strauss, 1997: Evaluation of probabilistic prediction systems. *Proceedings, ECMWF Workshop on Predictability, ECMWF*, 1–25. [Available from ECMWF, Shinfield Park, Reading, Berkshire RG2 9AX, United Kingdom].

I think that acknowledging the connection to forecast verification would provide additional support for using this approach.

More importantly, there are standard (perhaps many?) methods for assessing the flatness of the rank histograms. These methods take into account the number of verifying observations and ensemble size. The present work does not use any statistical methods to assess the flatness of the rank histogram and instead introduces admittedly ad hoc procedures to determine whether model performance is adequate. Using rigorous methods to assess the rank histograms would better quantify uncertainty and potentially strengthen and simplify the results. An important question which could be answered (currently unanswered) is whether *any* of the simulations have variability that is statistically consistent with observations. A classic reference on assessment of rank histograms is

Jolliffe, I. T., and C. Primo, 2008: Evaluating Rank Histograms Using Decompositions of the Chi-Square Test Statistic. *Mon. Wea. Rev.*, 136, 2133–2139, <https://doi.org/10.1175/2007MWR2219.1>.

There may be better, newer methods, too.

3. As noted in point 1, I did not find it completely clear what hypothesis is being tested in the current work or exactly what the conclusion is. Presumably, the overall goal is more accurate projections. This, of course, is impossible to assess since we don't know the future. Moreover, there is the familiar issue that past performance does not guarantee future performance, a point which I did not see mentioned. The present work emphasizes reductions (or lack of) in projection uncertainty in the performance-constrained ensemble. However, as noted in the text, these reduction might simply be due to removing models. Perhaps this hypothesis could be tested. That is, are the changes in uncertainty greater than would be expected if models were removed at random instead of based on historical performance?

Since we don't have observations of the future, a standard method of evaluating model weighting/subsetting schemes is the so-called perfect model approach where one model at a time is treated as "observations." By doing so, it can be checked whether a subsetting strategy based on historical performance actually gives more accurate projections. This method is widely used, including in the Brunner reference above and this one

Knutti, R., J. Sedláček, B. M. Sanderson, R. Lorenz, E. M. Fischer, and V. Eyring (2017), A climate model projection weighting scheme accounting for performance and interdependence, *Geophys. Res. Lett.*, 44, 1909–1918, doi:10.1002/2016GL072012.

Note that both works also consider model independence which might not be an issue here but could be considered and mentioned.

4. Figures 2–5 show constrained vs. full ensemble results and differences without statistical significance statements. I find it hard to interpret whether the results are "significant" or just "different." I think the sample size is fairly large (11 years x ensemble size) but variance tends to be more subject to sampling variability than means. I'm not sure that the ensemble-based error bars correctly capture the uncertainty since sampling variability goes up with reduced ensemble size (right?) and the error bars would tend to shrink?

Other comments.

I would find the figures easier to read if the captions and plots used the term standard deviation instead of temperature variability. Figures 2 and 3 do say "temperature variability defined as the standard deviation" but 4 and 5 don't. Seems like standard deviation would be more precise and concise.

Line 43. "However, it remains unclear whether climate models disagree in the direction and intensity of temperature variability changes because they do not adequately simulate temperature variability over certain regions, or because of other, more systematic model differences in future temperature variability behaviour. In this study, we address this question by constraining projections of temperature variability in a multi-model super ensemble of single model initial-condition large ensembles (SMILEs) to models that adequately represent observed regional variability under current climate conditions." This seems important but was not very clear to me how the question was being addressed. I think the idea (after reading the paper) is that if the performance-based ensemble doesn't have smaller uncertainty, then other differences (i.e., ones that do not result in variability biases during the historical period) are present.

Figure 1. What is the purpose of the time series shown? Are they mentioned in the text? I was unable to make out the very light red and blue shading. The caption does not say what the Above, Below, and Center values are. I think the gray lines are a 90% confidence interval but I could not follow the description (below) in the caption. In particular, if the lines illustrate the slope, the slope should be zero but I don't see zero being plotted. Perhaps say what the lines represent and give the details in the methods section instead of the caption. I didn't see any more details in the current methods section.

Fig. 1 caption: "Lines illustrate the rank histogram's slope, as the mean rank frequency over a centered 6-bin window for observations (coloured lines), and the 5-95th percentile perfect model range (the slopes of all ensemble members treated as if they were observations; gray dashed lines)."

"Crosses represent the frequency of minimum (0) and maximum (number of members) ranks for observations (colors), and for the 5-95th percentile perfect model range (gray)" Isn't the frequency of the observation rank already given by the bars?

Line 113-116. "we also include time series and rank frequency histograms for all regions and models where model performance based on spatially aggregated data can be assessed. Furthermore, time series for this spatially aggregated

evaluation also allow changes in potential biases over different periods to be assessed." I don't understand what it means to include time series or what the time series in question are. If this is a technical detail, perhaps keep move it to the methods section where it can explained more completely. Also repeated later.

Table 1 caption "total amount" -> "number"

Line 132 "struggling" -> "struggle"

Fig. 2. Starting at zero seems to waste considerable space and reduce readability. There is no objective measure here of whether projection values are different between the constrained and full ensembles and also whether uncertainty is reduced. This issue is especially unclear when only a few models are retained.

"Horizontal lines are plotted at both 1 degree (solid line) and 4 degrees (dashed line) of warming." Why? Don't the bars show the same information? Perhaps it would help to put the constrained and unconstrained values for the same warming level beside each other to aid comparison.

Line 206 "constraining the projections to best-performing models yields a decrease in model spread" Would selecting any subset of models (including randomly) yield a decrease in spread?

Line 207. "Less" -> fewer

Line 215. "These results indicate the performance constraining can indeed decrease the uncertainty in future variability projections" It seems likely that many criteria for removing models reduce spread.

Line 233. "temperature variability is significantly overestimated at the 1oC warming level." Compared to observations? Or something else?

Fig. 4. Is it possible to indicate statistical significance of the differences?

Fig. 4 caption and elsewhere. Is that is a degree symbol?

The difference of STD in Fig. 4I looks small but I'm not sure what is considered small.

Line 243. "This implies that the absolute value of the variability at 3oC is overestimated in the full ensemble, but the change in variability is underestimated, meaning that the interpretation of results could be biased in either direction depending on which metric is used." I find this hard to follow. What does "the absolute value of the variability" mean? STD is always positive? Which "interpretation of the results" is being referred to here?

Line 249 "it's" -> its

Fig. 5. Possible to add or comment on statistical significance? Especially for panel I.

Line 257. "Typically regions where the constrained ensemble includes many less models than the full ensemble are regions where model agreement is substantially changed." Is this obvious? If so, does it provide a meaningful conclusion? Also less -> fewer.

Line 260. "the spread of projections and hence the magnitude of the change can be limited using this method." Is this the goal? Is it a meaningful metric of success? This is a place where the using one model as observations might permit stronger results.

Is Fig. 6 mentioned in the text? Does Fig. 6 provide anything that's not in Figs. 4 and 5? Is a t-test the right test for a difference in variances? Or F-test? Though this is a difference in variances. Again, isn't comparing the full ensemble to one with fewer models a bit of a straw man? Also I cannot visually tell the difference between small values (white) and insignificant values (also white?). What does "general region" mean in the caption? Is there some smoothing?

Line 267. "This evaluation enables end-users of these ensembles to select the best fitting model for their region and time of interest" Is there evidence that such a strategy leads to more accurate projections in the perfect model (each model in turn as observations) setting?

Line 274. "they may, however, underestimate the projected temperature variability change" or they may not? Right?

Line 277. "providing an improved estimate" In what sense is the estimate improved?

Line 281. "In these regions we find that temperature variability itself is typically overestimated in present day climate; while the constrained ensemble provides a larger projected change in variability." It would be helpful to provide the reader with pointer to where these results appear. Or perhaps some summary graphic showing this relation?

Line 310. "Our attempt to constrain projections of future temperature variability change is an example of change in the climate system that cannot yet be foreseen with the best available knowledge." What does this mean?

Line 365. "date" -> data

Line 370. "Lastly, to allow a more finely resolved temporal analysis and to increase sample size we perform this evaluation on monthly mean temperature anomalies instead of seasonal averages." What evaluation? The rank-frequency evaluation or another diagnostic?

Line 384 "To illustrate how internal variability may affect rank frequencies" isn't the rank histogram itself assessing variability? Or do you mean finite sample size? Line 397 "To quantify when variability biases are present, we select the following thresholds." These seem arbitrary. It would be sensible to include some sort of significance testing of rank histograms. Line 408. "Note that the variability underestimation threshold of 8% is chosen to be rather conservative to account for the different ensemble size" Wouldn't it be more objective to use a standard significance test which accounts for ensemble size? Perhaps these are all points where a method that takes into account sample and ensemble size would be beneficial.

Line 385 "and" -> an

Line 428 "Lastly, we determine, per grid-cell, if a model simulates observed temperatures adequately when none of these biases are present." Is this saying when none of the biases in the previous paragraph are present, an additional procedure is applied to determine if a model simulation is adequate? If so what is that procedure?

Lines 430–433 are a repeat of lines 111–114? I still don't quite get what the time series refer to.

Line 459. "Significance for Figure 7 is calculated using a t-test approach for each domain." Fig. 7 is the domains?

(Remarks on code availability)

Reviewer #2

(Remarks to the Author)

Review for "Temperature variability projections remain uncertain after constraining them to best performing SMILEs"

I have read the manuscript and have several concerns, mainly about the scope of the study. The authors essentially use output from large ensembles of climate model simulations, evaluate whether or not they have a good representation of historical temperature variability, then assemble a "constrained ensemble" based on admittedly subjective criteria to evaluate how much removing models with poor representations of historical variability changes future projections. In general, they find that the answer is basically unchanged with or without their constraint.

While this is a worthy effort, I do not believe this constitutes a novel study that adds to our understanding of temperature variability in climate model simulations. The authors provide a detailed description of their results, and while I have no objection to their findings or methodology (though I am slightly confused by the ranking figure) I do not believe this paper constitutes a sufficient advance to be considered in Nature Communications.

Major Comments:

Figure 1: The shading is extremely difficult to see, which makes this plot difficult to interpret. I found the discussion of rank histograms slightly confusing and thought the authors should say more about what a "desirable" scenario would be for these histograms. Presumably it would be that the observations sit somewhere in the middle of the ensemble, but that's just my guess.

The discussion of detrended versus non-detrended data was interesting, but qualitative and speculative. I was also confused by lines 172-175, where the authors say "The higher agreement between non-detrended data and observations could arise from the fact that misrepresented forcing signals can counteract commonly-found variability overestimation biases" Wouldn't this be the opposite? I would think that higher agreement between non-detrended data and observations would arise from accurate representation of the forcing signal, rather than misrepresentation.

Minor Comments: Figure 6 never called or discussed?

(Remarks on code availability)

Version 1:

Reviewer comments:

Reviewer #1

(Remarks to the Author)

The authors have responded carefully to all my comments. I have one comment about a possible significance testing approach and a few minor comments.

The authors state in their response "Therefore, if the reviewer has any specific suggestions as to how to appropriately calculate significance for this study we are open to including them." I think that a permutation test is appropriate in this case, as it does not rely on distributional assumptions and is exact under the null hypothesis of exchangeability. Specifically, compute the difference in variances between the constrained and full ensembles. Then, repeatedly create random constrained ensembles (of the same size etc) by sampling without replacement from the full ensemble, and compute the same variance difference for each permutation. The p-value (two sided) is the fraction of permutations where the absolute value of the permuted variance difference exceeds that of the observed difference. I think that your calculation mentioned at line 380 is very close to this.

Minor comments

Line 46 define SMILE at first use in text (not at line 83)

Line 56 "with 60N"?

Line 57 "This is also a region of decreasing variability" what is "this"?

Line 72 "Therefore, a possible hypothesis is that future temperature variability depends on historical variability" I don't know what this means. I think you want to mention models. Perhaps, "Therefore, a possible hypothesis is that the ability of models to project future temperature variability depends on their ability to simulate historical variability"

Line 78 "In contrast, the alternative hypothesis is that future temperature variability is intrinsic to how the Earth system evolves over different regions." I don't know what this means.

Line 128 Delete comma

Line 153 Readers will recognize "perfectly flat" as a bit of a straw man since no tests require perfect flatness, and the usual tests take sample size into account. The usual chi-squared test does (I think) assume independence. You could drop the sentence starting at line 150 and simply say at line 154 "We tested the flatness of the rank histogram using a perfect-model rank range test which make no distributional assumptions and takes into account both ensemble size and serial correlation"

Line 166 and throughout "Criteria" is plural so Criterion 1?

Line 254 space before GFDL

Lines 337 and 338 em dashes instead of hyphens

(Remarks on code availability)

Reviewer #2

(Remarks to the Author)

In my original review, I expressed concerns about the novelty of this study and the importance of the results. The authors have done a better job at explaining why their study is different from the original 2021 study that used a similar approach to understand temperature variability in the CMIP5 models. I appreciate the new description of the two evaluation criterion, which helps clarifies what distinguishes this approach from the previous efforts to understand temperature variations in climate models.

The result that the constrained ensemble does not help projections of temperature variability converge is an interesting one, but in the authors' response they write a hypothesis that "future temperature variability is intrinsic to how the Earth system evolves over different regions, and therefore even constraining projections of future temperature variability change to models that best capture historical variability will not necessarily reduce future uncertainty if those models that capture historical variability adequate[ly] still exhibit diverging behaviors in the future." This is definitely supported by their results, but without an explanation of why the models disagree with respect to future climate change, it's not really satisfying to me as a result. I completely understand, and agree with, the authors' point that "uncertainty in our future projections that cannot just be improved by picking 'the good models'" but I think this isn't really a knowledge gap, but more of a philosophical problem in the climate modeling community.

In general the authors have addressed my comments thoroughly, which I appreciate given my somewhat critical review.

(Remarks on code availability)

**Response to the Reviewers' comments on manuscript NCOMMS-25-1462
“Temperature variability projections remain uncertain after constraining them
to best performing SMILES” by Suarez-Gutierrez & Maher.**

Please find enclosed our response to the Reviewers' remarks on our manuscript. Following their suggestions, we have revised our work by expanding the introduction and conclusions sections to clarify the goals and implications of the study and to better place our contribution in the context of prior work on performance- and independence-based constraining, highlighting the novelty of applying such methods to temperature variability. We have also strengthened the evaluation framework by introducing a new formally defined performance criteria based on whether observational rank histograms fall within perfect-model rank ranges, thereby providing a more rigorous and conceptually consistent evaluation benchmark. We have also updated the main figures to better showcase how our variability estimates compare to observations and to clarify the extent of the uncertainty reduction achieved by the performance constraining, completely changing the concept of Figure 7 (previously Figure 6) to compare future projections to the projections by the ‘best performing model’. We thank the reviewers for these suggestions and comments, which we believe have greatly improved our work and which we address individually more in depth, along with all other remarks, below (in blue).

Reviewer #1 (Remarks to the Author):

Summary and main comments

The authors evaluated historical simulations of seasonal (DJF and JJA) temperature variability in 11 large ensembles compared to observations using rank-frequency analysis (known as rank histograms in the ensemble forecasting literature). Models whose historical performance was judged adequate (by region and season) were included in a performance-constrained ensemble. Projections based on the full and performance-constrained ensemble were compared.

1. The introduction provides few (one?) references to application of performance-based constraints/subsetting to projections. There is an extensive literature on this topic, and I think it would be helpful to connect the current work to it. The other topics (importance of temperature variability and SMILES) are well introduced. A arbitrarily chosen example is

Brunner, L., Pendergrass, A. G., Lehner, F., Merrifield, A. L., Lorenz, R., and Knutti, R.: Reduced global warming from CMIP6 projections when weighting models by performance and independence, *Earth Syst. Dynam.*, 11, 995–1012, <https://doi.org/10.5194/esd-11-995-2020>, 2020.

This would also provide an opportunity to give the reader a clear idea of what has been done before in this direction and how this work improves upon previous efforts. It may be the case that model weighting/subsetting has not been applied to questions of variability changes, and if so, pointing this out would highlight the value of the work. Or perhaps other approaches suffer from limitations that are addressed here. At present, I found the limitations of past work/goal of present work statement not very

clear (line 43): "However, it remains unclear whether climate models disagree in the direction and intensity of temperature variability changes because they do not adequately simulate temperature variability over certain regions, or because of other, more systematic model differences in future temperature variability behaviour. In this study, we address this question by constraining projections of temperature variability in a multi-model super ensemble of single model initial-condition large ensembles (SMILEs) to models that adequately represent observed regional variability under current climate conditions."

We thank the reviewer for highlighting this. We have now expanded the introduction to expand on the previous work on performance and independence weighting, adding a new paragraph on line 88. We have also clarified that such performance subsampling approaches have not been applied to temperature variability before and our study is the first to do so (Line 105).

We have also updated line 43, now line 69 for clarity as follows:

"It remains however unclear whether climate models disagree on the direction and intensity of temperature variability changes because they do not adequately simulate present-day temperature variability correctly, or whether they disagree due to intrinsic model differences in how the drivers of temperature variability change in a future climate. Therefore, a possible hypothesis is that future temperature variability depends on historical variability. If this were true, subsampling models based on their historical performance in capturing observed variability would yield reduced uncertainty in future projections. This means that by constraining projections to models that best match observations and agree on their historical variability representation, the range of future variability change projections across best-performing models would be reduced, and thus be less uncertain. In contrast, the alternative hypothesis is that future temperature variability is intrinsic to how the Earth system evolves over different regions. If this were true, subsampling models based on their historical performance would not necessarily reduce future uncertainty, as long as those best-performing models still exhibit diverging behaviours in their representation of future temperature variability changes. In this study, we address these questions by, first evaluating in a multi-model super ensemble of single model initial-condition large ensembles (SMILEs) which models can adequately represent observed regional variability under current climate conditions. Second, we subsample the super ensemble to those adequate models, per season and region, and compute constrained future projections for regional temperature variability. Third, we compare the full ensemble with this subset of the ensemble to determine if the uncertainty in our projections is lowered by constraining them to only models that adequately capture historical variability."

2. Rank-frequency analysis or more commonly rank histograms are a standard metric for assessing the reliability of ensemble forecasts. Classical references are

Anderson, J. L., 1996: A method for producing and evaluating probabilistic forecasts from ensemble model integrations. *J. Climate*, 9, 1518–1530.

Hamill, T. M., and S. J. Colucci, 1997: Verification of Eta–RSM Short-Range Ensemble Forecasts. *Mon. Wea. Rev.*, 125, 1312–1327, [https://doi.org/10.1175/1520-0493\(1997\)125<1312:VOERSR>2.0.CO;2](https://doi.org/10.1175/1520-0493(1997)125<1312:VOERSR>2.0.CO;2).

Talagrand, O., R. Vautard, and B. Strauss, 1997: Evaluation of probabilistic prediction systems. Proceedings, ECMWF Workshop on Predictability, ECMWF, 1–25. [Available from ECMWF, Shinfield Park, Reading, Berkshire RG2 9AX, United Kingdom.].

I think that acknowledging the connection to forecast verification would provide additional support for using this approach. More importantly, there are standard (perhaps many?) methods for assessing the flatness of the rank histograms. These methods take into account the number of verifying observations and ensemble size. The present work does not use any statistical methods to assess the flatness of the rank histogram and instead introduces admittedly ad hoc procedures to determine whether model performance is adequate. Using rigorous methods to assess the rank histograms would better quantify uncertainty and potentially strengthen and simplify the results. An important question which could be answered (currently unanswered) is whether *any* of the simulations have variability that is statistically consistent with observations. A classic reference on assessment of rank histograms is

Jolliffe, I. T., and C. Primo, 2008: Evaluating Rank Histograms Using Decompositions of the Chi-Square Test Statistic. *Mon. Wea. Rev.*, 136, 2133–2139, <https://doi.org/10.1175/2007MWR2219.1>.

There may be better, newer methods, too.

We thank the reviewer for highlighting the connection to the forecast verification literature and for providing relevant references, which have been added to the revised manuscript.

Regarding the use of standard statistical methods to assess the flatness of rank histograms, or whether any simulations exhibit variability that is statistically consistent with observations, we appreciate the reviewer's suggestion. We agree with the reviewer that applying formal statistical tests could, in principle, yield a more rigorous framework for evaluating histogram flatness. However, the relatively short length of the observational record means that the influence of internal variability cannot be robustly sampled in these timescales for the metrics considered here, therefore the observed variability estimates derived from this record would not necessarily reflect the full range of real-world variability, and may not be a suitable standard to evaluate the models. Therefore, tests assessing the statistical similarity to observations or flatness of rank histograms directly may lead to misleading conclusions about model performance, in a similar way than assessing model performance based on only one or a few realizations yields a less robust assessment than using a large ensemble of simulations.

This lack of robust variability sampling from the finite and relatively short observational record, combined with the fact that we are comparing observations to uninitialized, free-running model simulations means that, with the sample size available, we would for example not expect rank histograms to be perfectly flat, even for perfectly performing models. This is highlighted in our analysis by including a *perfect-model rank histogram range* (e.g., in Fig. A below, now Fig. 9 in the revised manuscript). This range is constructed by treating each ensemble member in each SMILE as if it were observations and calculating the resulting spread of model rank histograms, which can be seen in the example figures below (gray dashed lines and crosses in left panels in example Fig. A). This approach provides a relatively wide distribution of rank histograms that reflect perfect-model behaviour, and are indeed not perfectly

flat (such as almost twice as high perfect-model upper bound for 0 rank frequency than for $n+1$ rank frequency for L7 region in Fig. A, for example). This can occur either systematically because the simulated temperature distribution may be skewed or non-normal, or because the record length is too short to sufficiently sample rank variability, or both.

We recognize however that, while an statistical assessment of observations rank histogram flatness may not be the most adequate approach to assess model performance in this case, our evaluation framework could still be improved from our original approach to incorporate these principles. Therefore, we have substantially expanded our evaluation framework to include a new formally defined evaluation criteria: that the observations rank histogram fits within the perfect-model rank histogram range. The perfect-model rank range delineates a range of deviations from flatness that would be plausible for a model that perfectly captures observations for the metrics and record length considered here. Therefore, we argue that as long as the observations rank histogram is within the perfect-model rank range, the model in question adequately simulates the variability in observations for that particular region, even if the histogram of the observations itself is not flat. Considering the limitations of the observational record discussed above, we believe that using this perfect-model rank range provides a more appropriate benchmark in this context and greatly improves our evaluation framework.

While the perfect-model rank range was already included in our original analysis as a visual element, it is now also used as an additional evaluation criteria. This new evaluation criteria, now Criteria 1, is performed over spatially averaged metrics, and is complemented by a slightly revised version of our original grid-cell level evaluation criteria, now Criteria 2, to account for biases at the grid-cell level that may be smoothed or compensated in this spatial average. Therefore, the two evaluation criteria used in our framework are now as follows:

1. **Regional perfect-model rank range performance:** observations rank histogram of the spatially averaged temperature for a given region must lie within the perfect-model rank range across all ranks.

The number of total ranks depends on the ensemble size, ranging from 0 to n , with n being the number of ensemble members. Therefore, to minimize the effect of comparing ensembles of different sizes and potentially spurious effects resulting from the insufficient rank variability sampling due to the finite sample size, both model and observations rank frequencies are shown as the running mean across individual rank frequencies over an $n/5$ rank window (for $n=50$ members, 10-rank running average; dashed gray and solid color lines in rank histograms in Fig. A) for ranks 1 to $n-1$, and as the actual rank frequency for ranks 0 and n (gray and color crosses in Fig. A). We allow observations rank running-mean to exhibit a maximum deviation outside of this perfect-model range of 10% (e.g., for an upper limit in the smoothed perfect-model rank range of 3% for a particular rank window, a maximum observations rank frequency of 3.3% would be accepted for adequate performance). The fulfilment of this criteria for all rank windows is marked by a green star in the histogram panels (e.g., Fig. A, Fig.1 in the revised manuscript), while inadequate performance is marked by a red cross.

2. **Threshold-based grid-cell level performance:** at least 50% of the region's grid-cells must be unbiased, meaning, at the grid-cell level, a.) observations do not cluster excessively within the central percentiles of the ensemble (observations occur with more than 80% frequency within the central 75th percentile ensemble bounds, indicative of variability overestimation bias) and b.) fall outside the ensemble range too frequently (observations occur with more than 8% frequency outside of the ensemble spread, indicative of variability underestimation bias).

Criteria 2. has been updated from our original framework to account for 50% instead of 60% of unbiased grid cells in a region. This slightly relaxed threshold ensures the condition is not overly restrictive while still maintaining a robust standard for model adequacy. Only when both criteria are fulfilled, we consider the model in question offers an adequate performance and is selected to be part of the constrained ensemble for the particular region and season.

The addition of Criteria 1. makes our evaluation framework more restrictive than our original approach, with a 22% reduction in total number of regions being adequately captured (calculated maintaining the original threshold in Criteria 2 of 60% of unbiased grid-cells). Relaxing this threshold to 50% unbiased grid-cells yields approximately the same number of total regions being adequately captured as in our original framework (i.e., 354 originally, 353 now). However, even though the total number of adequately captured regions remains somewhat stable, this additional criteria affects the number of adequate regions per model, in some cases substantially (e.g., drop from 37 to 28 adequate regions for ACCESS, increase from 21 to 33 regions for MPI-GE5). The full account of the updated evaluation results can be found summarized in Table 1 for detrended data, and additionally in Table 2 for non-detrended data in the revised manuscript, as well as in the revised supplementary information files for all SMILEs.

This revised evaluation framework improves on our original approach in two key ways. First, by explicitly assessing the agreement between the observed rank histogram and the perfect-model rank range, incorporating the key aspects of the probabilistic verification framework suggested by the reviewer in a suitable way, accounting for sample limitations and the potential effect of limited internal variability sampling in rank histogram flatness. Second, the revised framework improves our assessment by ensuring that the full rank distribution and the effect of ensemble size is considered in our evaluation (Criteria 1.), while including the grid-cell level performance in a computationally efficient way (Criteria 2.). Applying Criteria 1. at the grid-cell level (calculating the perfect-model rank range assessment for each member of each SMILE for each grid-cell) would potentially be the more rigorous alternative to Criteria 2. However, we believe this would entail excessive computational costs and limit the applicability of our framework. Therefore, together these two complementary criteria provide a more comprehensive and conceptually consistent assessment of model performance.

We have revised the manuscript to reflect this expansion of the evaluation framework to explicitly incorporate the agreement between the observed rank histogram and the perfect-model rank range as a

performance metric and the formally defined evaluation criteria, both for the introduction (lines 126-132), results (lines 141-219) and methods sections (lines 514-625).

Figure A: Example time series and rank histogram evaluation for MIROC6 against GISTEMPv4 observations. Time series and rank histograms show the evaluation of spatially averaged temperatures for selected regions (Criteria 1). Time series show the ensemble maximum and minimum (color lines) and central 75th percentile ensemble spread (shading) against observations (dots). Rank histograms show observations rank frequency aggregated over a 3-rank window (bars), the running mean rank frequency over a centered $n/5$ rank window (lines; for 1 to $n-1$ ranks) and the absolute frequencies of rank 0 and n (crosses), with n the number of ensemble members, for observations (color) and perfect model rank range (gray). If observations rank frequency is within perfect-model range for all rank windows, Criteria 1 is met and this is highlighted by a green star at the top right; if not, by a red cross. Percentages at the top left show the frequency of regionally averaged observations occurring above (red) or below (blue) ensemble limits, or clustering within the central 75th percentile range (gray), analogous to the grid-cell evaluation in Criteria 2. Bottom maps show the grid-cell evaluation of temperature variability globally (Criteria 2). Gray hatching (variability overestimation bias) shows observations clustering within the central 75th percentile with more than 80% frequency. Red and blue shading (variability underestimation bias) shows observations falling beyond the ensemble maximum or minimum, respectively, with more than 8% frequency. Dotted areas represent ocean grid cells or where observations are missing and therefore excluded. Left column shows results for non-detrended data, right column shows analogous results for detrended data. Adapted from Fig. 9 in the revised manuscript. See Methods for further details and Supporting Information for results for all 11 SMILES.

3. As noted in point 1, I did not find it completely clear what hypothesis is being tested in the current work or exactly what the conclusion is. Presumably, the overall goal is more accurate projections. This, of course, is impossible to assess since we don't know the future. Moreover, there is the familiar issue that past performance does not guarantee future performance, a point which I did not see mentioned. The present work emphasizes reductions (or lack of) in projection uncertainty in the

performance-constrained ensemble. However, as noted in the text, these reduction might simply be due to removing models. Perhaps this hypothesis could be tested. That is, are the changes in uncertainty greater than would be expected if models were removed at random instead of based on historical performance?

We thank the reviewer for bringing these points to our attention. Indeed, the goal of our study is to reduce uncertainty in projections of temperature variability and its change. To clarify this, we now clearly state the specific aims of our work (lines 132-138 of the introduction), which are threefold :

- (i) Evaluate how well each SMILE captures historical temperature variability in both summer and winter seasons over individual regions.
- (ii) Assess model agreement in the projections of temperature variability across all SMILEs for both the sign of the change and the spread of the ensemble
- (iii) Constrain the projections using the SMILEs that perform well over the historical period, and assess whether this provides an increase in model agreement for future projections.

We have also restructured the introduction and discussion sections to present the goal of our work more clearly (e.g., lines 69-87 and 123-131).

Furthermore, we agree with the reviewer that adequate past performance does indeed not guarantee future performance, although studies show that in some cases it can, such as in Brunner et al 2020. We have expanded the introduction to clearly reflect these issues in lines 88 and 92. We have also expanded our analysis and the accompanying discussion of the results where relevant to more clearly address to what extent this uncertainty reduction is achieved over the different metrics considered (e.g., paragraphs on lines 297 and 365)

Regarding the reviewer's comment on whether the model spread and thus uncertainty in the projections would decrease simply by the fact of removing models from the sample randomly, we thank the reviewer for bringing this up. We have tested the hypothesis of randomly removing models. To do this we repeated the analysis in Figures 2 and 3 by creating thousand 6-model constrained ensembles randomly rather than by using our constraining method. We find that a decrease in the model spread is not guaranteed by design in this case. A decrease in the spread occurs only if the models at the ends of the spread are randomly removed, but not otherwise.

We note that the errorbars on Figures 2-3 show actually the end members of the model spread. We have one estimate for each model of variability (STD over the ensemble) as such these errorbars only decrease if the constraint removes the end member. This was not clear in the original text and this has been clarified in the figure caption:

“Errorbars show the full model spread (i.e. the end members of each ensemble), with the fatter errorbars highlighting the 25th and 75th percentiles.

Lastly, to further clarify to what extent our approach reduces uncertainty in model projections, we have also expanded the results and discussion sections (e.g., paragraphs on lines 353, 365, 443) and added a new section, Conclusions and Implications (lines 458-484), that clearly states the conclusions emerging from our work, which are as follows:

“(i) We provide a comprehensive evaluation of the state-of-the-art SMILEs ability to represent the historical summer and winter temperature variability in observations. We identify CESM-LE and CESM2-LE as the SMILEs that provide the best representation of isolated temperature variability as well as of both temperature variability and forced change, with GFDL-SPEAR-MED and MPI-GE5 as close third in each category respectively. This multi-model evaluation across all available CMIP and CMIP6 SMILEs provides a basis for model selection for the assessment of temperature variability and extremes, which cannot be correctly simulated if the underlying variability is incorrect.

(ii) Our evaluation also shows that some regions are systematically not well represented such as the Southern Ocean (O9), the Indian Peninsula (L20), and Northern Australia (L22), as well as the Amazon basin (L6), Northern (L12) and Eastern Africa (L14), and Southern Australia (L23), all specifically in the local summer season. We also find that certain areas of the world are systematically well captured by most models, especially in the local winter season, namely North America (L1-L4), Southern South America (L8), Europe and Northern Africa (L9-L11) and North and Central Asia (L16-L19). Finally, we conclude that local winter seasons are generally simulated better by most models more often than local summer seasons.

(iii) Similar to the IPCC, we find low model agreement on the sign of the change over large parts of the Earth’s surface and large model disagreement on the magnitude of the projected change under future warming, particularly over the land surface and the tropical Pacific Ocean. This implies that we cannot afford to move away from multi-model ensembles that sufficiently capture the uncertainty in our future projections.

(iv) The constrained ensemble decreases model spread over some regions and gives a lower range of projected futures, providing a smaller range of potential future projections and greater model agreement, particularly in South America, India, Australia & South-East Asia, the west Pacific Ocean and Southern Ocean in DJF and north South America, Africa, parts of Australia and the Southern Ocean in JJA. However, the constraint does not substantially increase model agreement on the sign of the projected change.”

Since we don't have observations of the future, a standard method of evaluating model weighting/subsetting schemes is the so-called perfect model approach where one model at a time is treated as "observations." By doing so, it can be checked whether a subsetting strategy based on historical performance actually gives more accurate projections. This method is widely used, including in the Brunner reference above and this one

Knutti, R., J. Sedláček, B. M. Sanderson, R. Lorenz, E. M. Fischer, and V. Eyring (2017), A climate model projection weighting scheme accounting for performance and interdependence, *Geophys. Res. Lett.*, 44, 1909–1918, doi:10.1002/2016GL072012.

Note that both works also consider model independence which might not be an issue here but could be considered and mentioned.

We appreciate the reviewers idea to use a perfect model approach considering each individual ensemble member as if it were observations. We agree that this would be a valuable exercise and well in line with the perfect model rank range analysis we introduce in the evaluation section of this manuscript. A test as proposed would answer the question of whether future variability depends on historical variability, specifically for each model across those considered, which expands on our goal of reducing uncertainty across temperature variability projections. We agree that this would be an interesting exercise and an additional layer of rigour to the analysis proposed here. While we recognize the value of performing this additional check, carrying out this additional perfect-model assessment treating each member of each one of all 11 SMILEs as observations and redoing the evaluation and constrained variability projections analysis excluding the respective SMILE would increase the length and complexity of manuscript substantially. To keep the analysis presented here as concise yet robust as possible, we have included an adapted version of this perfect-model projections test suggested by the reviewer based on CESM2-LE alone. Removing CESM2-LE, one of the models closest to observations particularly over the most poorly represented regions, we recalculate the full and constrained ensemble for future projections and assess how closely it resembles the CESM2-LE future projections focusing on key, poorly captured areas, and distinguishing between the cases where CESM2-LE adequately captures observations from those where it does not.

We explore this comparison of the full vs. constrained projections against the projections of the 'good performing model' CESM2 below (Figs. B and C, replicating Figs 2 and 3 in the revised manuscript, respectively, but now excluding CESM2). These results highlight how the *new* constrained (excluding CESM2) and the full ensemble (also excluding CESM2) compare with CESM2 (blue stars for regions and sessions when CESM2 is a good performer, red when it is not). These Figures and related discussion have been included to the paper (Figs. 10 and 11, lines 664-686)

Our main conclusion is that sometimes the constrained ensemble of adequately performing models is closer to CESM2 when it is a good performer, and sometimes it isn't. Likely, this is because in some cases future variability is more closely linked to past variability performance, versus in other cases climatic and environmental changes intrinsic to each model may affect the temperature variability distribution differently across different models, even for those that show a common and adequate historical representation. Disentangling the drivers of these diverging behaviours for future variability change across historically adequate performing models would be an interesting potential follow up work. This would entail a careful consideration of the relevant processes involved in temperature variability change across different regions and rigorous process-based evaluation. However, albeit interesting, we believe this is out of scope of this manuscript at this stage.

Figure B. DJF Comparison of ‘new’ constrained ensemble without CESM2 against CESM2 projections (stars). Blue stars mark regions for which CESM2 offers an adequate representation of historical variability, red stars represent where it does not. Same as Fig. 10 in the revised manuscript.

Figure C. JJA Comparison of ‘new’ constrained ensemble without CESM2 against CESM2 projections (stars). Blue stars mark regions for which CESM2 offers an adequate representation of historical variability, red stars represent where it does not. Same as Fig. 11 in the revised manuscript.

Furthermore, we have also completely revised Figure 6 (now Figure 7) to expand this new assessment over key, poorly modeled areas, and expanded the related discussion on line 353 as follows:

“While we cannot assess whether the future projections are more realistic in the constrained ensemble compared to the full ensemble, as we do not have observations of the future, we can compare the projections with those from the most realistic model over these poorly modelled regions in the historical period. CESM2 is one of the two models that best represent Australia & South-East Asia (L21-23), South America (L6-8), and Africa (L12-16; the other being CESM-LE), with 15 “good performances” out of 22 across both regions and seasons. By using this model as a proxy for possible future observations, we can compare the differences at 3 °C of warming between the full and constrained ensemble, the constrained ensemble (excluding CESM2), and CESM2 (Fig. 7, see Figs. 10 and 11 for a spatially aggregated comparison across all regions). We find that, except for South America (where CESM2-LE fails to capture historical variability according to our evaluation for L6 and L7) and Africa in DJF the constrained estimate is closer to CESM2 (our “good model”) than the full ensemble, tentatively suggesting that our future projections might be more realistic in the constrained ensemble compared to the full ensemble. ”

4. Figures 2–5 show constrained vs. full ensemble results and differences without statistical significance statements. I find it hard to interpret whether the results are "significant" or just "different." I think the sample size is fairly large (11 years x ensemble size) but variance tends to be more subject to sampling variability than means. I'm not sure that the ensemble-based error bars correctly capture the uncertainty since sampling variability goes up with reduced ensemble size (right?) and the error bars would tend to shrink?

Regarding the remark on the statistical significance of these results, we thank the reviewer for the suggestion, but we are not sure it is feasible to perform such tests considering our sample. For typically used tests, such as a t-test or a f-test, they are not appropriate to use on these data, as the full and constrained ensembles are not independent samples, and would violate the assumptions of both statistical tests. We have not found a test that would appropriately assess this statistical significance without violating the core assumptions of said test. Therefore, if the reviewer has any specific suggestions as to how to appropriately calculate significance for this study we are open to including them. To overcome these issues and the lack of suitable statistical significance testing, we include the bottom two panels in Figure 4 and 5s. The second last row shows the percentage of models that agree on the sign of the change and the last row shows the standard deviation (STD) across models, effectively illustrating the spread of the model projections. We note that we changed the difference of this last row to be a percentage change, which is easier to interpret visually. We propose these two bottom rows of figures to assess the significance of our results as an effective alternative to potentially flawed statistical tests that may lead to misleading conclusions on this significance of our results. While one might assume that the STD would decrease as we remove models, we have tested this assumption and find that the magnitude of the STD is dependent on which of the models are removed and it can increase or decrease. We had added discussion on this point on line 378.

“Typically, regions where model agreement on the sign of the change is substantially improved are regions where the constrained ensemble includes many fewer models than the full ensemble. Note, however, that this is not a predetermined result. We tested the change in the standard deviation (understood as the uncertainty in the magnitude of the change) in a constrained ensemble of either 4 or 6 randomly selected models by sampling 1000 times, and find the standard deviation decreased approximately 2/3 of the time and increased 1/3 of the time. ”

We have also tested the hypothesis of randomly removing models just decreasing the model spread by design and find this is not predetermined. Text has been added on line 298:

“The errorbars themselves show the multi-model ensemble spread, and are thus determined by the end members of the multi-model ensemble. The interquartile range of the multi-model ensemble is shown in the wider bars. By design, the errorbars only decrease when the end members of the ensemble are removed. Therefore, only if our evaluation proves the more extreme model members to not perform adequately, and thus removes them from the constrained ensemble, does the uncertainty in the projections decrease. This becomes more likely to happen by chance with the removal of more models, but is not a predetermined nor expected outcome.”

Other comments.

I would find the figures easier to read if the captions and plots used the term standard deviation instead of temperature variability. Figures 2 and 3 do say "temperature variability defined as the standard deviation" but 4 and 5 don't. Seems like standard deviation would be more precise and concise.

We have added this definition to the captions of Figures 4 and 5. We wish to keep temperature variability defined as the standard deviation, as it is important that the reader knows that what we are discussing is this variability. Using only the standard division has the possibility of the link between standard deviation and temperature variability being missed.

Line 43. "However, it remains unclear whether climate models disagree in the direction and intensity of temperature variability changes because they do not adequately simulate temperature variability over certain regions, or because of other, more systematic model differences in future temperature variability behaviour. In this study, we address this question by constraining projections of temperature variability in a multi-model super ensemble of single model initial-condition large ensembles (SMILEs) to models that adequately represent observed regional variability under current climate conditions." This seems important but was not very clear to me how the question was being addressed. I think the idea (after reading the paper) is that if the performance-based ensemble doesn't have smaller uncertainty, then other differences (i.e., ones that do not result in variability biases during the historical period) are present.

Yes, the reviewer's interpretation is correct. We have rewritten this paragraph to more clearly reflect this reasoning (starting in Line 69), which now reads as:

“It remains unclear whether climate models disagree on the direction and intensity of temperature variability changes because they do not adequately simulate present-day temperature variability correctly, or whether they disagree due to intrinsic model differences in how the drivers of temperature variability change in a future climate. Therefore, a possible hypothesis is that future temperature variability depends on historical variability. If this were true, subsampling models based on their historical performance in capturing observed variability would yield reduced uncertainty in future projections. This means, by constraining projections to models that best match observations and agree on their historical variability representation, the range of future variability change projections across best-performing models would be reduced, and thus be less uncertain. In contrast, the alternative hypothesis is that future temperature variability is intrinsic to how the Earth system evolves over different regions. If this were true, subsampling models based on their historical performance would not necessarily reduce future uncertainty, as long as those best-performing models still exhibit diverging behaviours in their representation of future temperature variability changes. ”

We agree with the reviewer’s interpretation that, in cases where the performance-constrained ensemble does not have reduced uncertainty, future differences could be driven by different evolutions of the Earth system in these regions intrinsic to each model; while this intrinsic model behaviour may not affect historical performance as discussed above. As an hypothetical example, over areas such as the Amazon basin this could be caused by changes to the Amazon ecosystem not strictly driven by local temperatures (e.g., from changes in precipitation patterns to drastic forest composition changes or forest die-back), that could cause this ecosystem and its associated temperature variability to evolve in different ways in different models, even though today’s climate may be similar and well captured in the models. Disentangling the drivers of diverging temperature variability behaviour across different, potentially adequately performing models over different regions would be an interesting yet complex exercise. This assessment would need to be carefully performed considering process-based evaluation approaches across relevant metrics guided by regional expert knowledge of both climatic and environmental change over key hotspot regions of variability change uncertainty. We believe this would make for an interesting and extremely relevant follow-up paper, or several papers, but is at this stage beyond the scope of our current manuscript.

Figure 1. What is the purpose of the time series shown? Are they mentioned in the text? I was unable to make out the very light red and blue shading. The caption does not say what the Above, Below, and Center values are. I think the gray lines are a 90% confidence interval but I could not follow the description (below) in the caption. In particular, if the lines illustrate the slope, the slope should be zero but I don’t see zero being plotted. Perhaps say what the lines represent and give the details in the methods section instead of the caption. I didn’t see any more details in the current methods section.

Fig. 1 caption: "Lines illustrate the rank histogram’s slope, as the mean rank frequency over a centered 6-bin window for observations (coloured lines), and the 5-95th percentile perfect model range (the slopes of all ensemble members treated as if they were observations; gray dashed lines)."

"Crosses represent the frequency of minimum (0) and maximum (number of members) ranks for observations (colors), and for the 5-95th percentile perfect model range (gray)" Isn't the frequency of the observation rank already given by the bars?

We thank the reviewer for highlighting these unclear points. The time series shown reflect the base for the spatially aggregated rank-frequency evaluation, now Criteria 1. We believe including the time series alongside the rank histograms is useful to help readers not familiar with rank histograms to understand how they are constructed and what different shapes translate to in terms of potential biases, therefore we choose to keep both visual elements in the main manuscript. We have however expanded the text clearly stating the meaning and relevance of these figures in the main results section (lines XX). Furthermore we have revised the format of the map figures to make the hatching more clearly readable.

We have also extended the figure captions for this type of figure to clearly describe all visual elements through the paper (i.e., Figs. 1, 9 and Supporting Information). To clarify the elements brought up in the remark above specifically:

- The 'Above, Below, Center' frequencies mark the frequencies with which spatially averaged observations fall within the rank windows assessed in the grid-cell based evaluation, now Criteria 2 (namely above the ensemble maxima or rank n , with n being the number of ensemble members; below the ensemble minimum or rank 0; and within the central rank window marked by the central 75th percentile ensemble bounds).
- The lines in the rank histograms mark the running mean of the rank frequencies averaged over a $n/5$ rank window. The color lines illustrate this running mean for observations (for which rank frequencies are also shown in the bars accumulated over a 3-rank bin), while the gray lines mark the perfect-model rank range in this metric as described above.
- The crosses mark the absolute rank frequencies for rank 0 and rank n for observations (in color) and for the perfect-model range (in gray). We choose to highlight this individual frequencies for minimum and maximum rank since they reflect the variability underestimation bias highlighted in Criteria 2 (i.e., 'Above, Below' frequencies in rank histograms, Red and Blue hatching in maps).

Lastly, since we have incorporated the perfect-model rank range into our valuation framework explicitly, for clarity we have modified the visual elements to illustrate the full range used and not the 5-95th percentile range as in the original manuscript.

Line 113-116. "we also include time series and rank frequency histograms for all regions and models where model performance based on spatially aggregated data can be assessed. Furthermore, time series for this spatially aggregated evaluation also allow changes in potential biases over different periods to be assessed." I don't understand what it means to include time series or what the time series in question are. If this is a technical detail, perhaps keep move it to the methods section where it can explained more completely. Also repeated later.

We thank the reviewer for highlighting this, which has now been clarified. The time series in question now reflect Criteria 1 of our evaluation and are introduced with more detail in the results and methods section, as well as in the supporting information where the time series and rank histograms that serve as based for the spatially aggregate evaluation in Criteria 1, and the maps for the grid-cell evaluation that serves as based for Criteria 2 can be found for all 11 SMILEs across all regions and seasons considered.

Table 1 caption "total amount" -> "number"

This has been changed as suggested.

Line 132 "struggling" -> "struggle"

This has been changed as suggested.

Fig. 2. Starting at zero seems to waste considerable space and reduce readability. There is no objective measure here of whether projection values are different between the constrained and full ensembles and also whether uncertainty is reduced. This issue is especially unclear when only a few models are retained.

Thanks for the suggestion, we have changed the Figure as suggested to be centred around the full ensemble at 1°C of warming. We have also updated the text so the errorbars are more easily understood, see caption:

“Error bars show the full model spread (i.e. the end members of each ensemble), with the fatter error bars highlighting the 25th and 75th percentiles.”

"Horizontal lines are plotted at both 1 degree (solid line) and 4 degrees (dashed line) of warming." Why? Don't the bars show the same information? Perhaps it would help to put the constrained and unconstrained values for the same warming level beside each other to aid comparison.

Thanks for the suggestion, we have removed the horizontal lines and put the constrained and unconstrained next to each other to aid comparison.

Line 206 "constraining the projections to best-performing models yields a decrease in model spread" Would selecting any subset of models (including randomly) yield a decrease in spread?

As discussed above, it would not. We have tested this and have elaborated on this in the paper:

Line 298: “The errorbars themselves show the multi-model ensemble spread, and are thus determined by the end members of the multi-model ensemble. The interquartile range of the multi-model ensemble is shown in the wider bars. By design, the errorbars only decrease when the end members of the ensemble are removed. Therefore, only if our evaluation proves the more extreme model members to not perform adequately, and thus removes them from the constrained ensemble, does the uncertainty in the projections decrease. This becomes more likely to happen by chance with the removal of more models, but is not a predetermined nor expected outcome.”

Line 378: "Typically, regions where model agreement on the sign of the change is substantially improved are regions where the constrained ensemble includes many fewer models than the full ensemble. Note, however, that this is not a predetermined result. We tested the change in the standard deviation (understood as the uncertainty in the magnitude of the change) in a constrained ensemble of either 4 or 6 randomly selected models by sampling 1000 times, and find the standard deviation decreased approximately 2/3 of the time and increased 1/3 of the time. "

Line 207. "Less" -> fewer

This has been changed as suggested.

Line 215. "These results indicate the performance constraining can indeed decrease the uncertainty in future variability projections" It seems likely that many criteria for removing models reduce spread.

As discussed above, this is generally not the case - see added text and response above to the comment on line 206.

Line 233. "temperature variability is significantly overestimated at the 1oC warming level." Compared to observations? Or something else?

We thank the reviewer for bringing this unclear statement to our attention. We have now added a new Figure 6 that compares the model ensembles with observations. We have also included the following text to explain this on line 323:

"This highlights the improved representation of the present day variability, as the lower variability in the constrained ensemble over land is in better agreement with observations, except for again in the northern hemisphere extratropics in DJF (Fig. 6). We note that the variability over the ocean shows little improvement in the constrained ensemble, and tends to be overestimated in DJF and underestimated in JJA. "

We have also updated line 233 (now line 339) to add "compared to observations".

Fig. 4. Is it possible to indicate statistical significance of the differences?

As highlighted in the major comments section above, this is not possible due to the lack of independence between the full and constrained ensemble samples that violate the assumptions in standard statistical tests.

Fig. 4 caption and elsewhere. Is that is a degree symbol?

We thank the reviewer for catching this typo, which seems to have emerged from an incorrect expression in LaTeX. We have modified this to show the correct degree symbol through the manuscript.

The difference of STD in Fig. 4l looks small but I'm not sure what is considered small.

We have changed this to be a percentage of the STD itself in the full ensemble for an easier visual assessment of how big the differences really are.

Line 243. "This implies that the absolute value of the variability at 3oC is overestimated in the full ensemble, but the change in variability is underestimated, meaning that the interpretation of results could be biased in either direction depending on which metric is used." I find this hard to follow. What does "the absolute value of the variability" mean? STD is always positive? Which "interpretation of the results" is being referred to here?

We have rephrased this to say 'This implies that the absolute value of the variability at 3°C is overestimated in the full ensemble, but the change in variability may be underestimated, meaning that the interpretation of results could be biased in either direction depending on whether one computes the actual value of future variability or the value of the future change in variability.'

Line 249 "it's" -> its

This has been changed as suggested.

Fig. 5. Possible to add or comment on statistical significance? Especially for panel 1.

As highlighted in the major comments section, this is not possible due to the lack of independence between the full and constrained ensemble that violate the assumptions in standard statistical tests.

Line 257. "Typically regions where the constrained ensemble includes many less models than the full ensemble are regions where model agreement is substantially changed." Is this obvious? If so, does it provide a meaningful conclusion? Also less -> fewer.

This is not obvious, but we have now elaborated on the regions, which reflect regions already discussed as those that include many fewer models in the constrained ensemble. Less has been changed to fewer.

Line 260. "the spread of projections and hence the magnitude of the change can be limited using this method." Is this the goal? Is it a meaningful metric of success? This is a place where the using one model as observations might permit stronger results.

We thank the reviewer for highlighting this statement, which was somewhat imprecise. Reducing the magnitude of the variability change is not the goal of our work. Instead, our goal is to reduce uncertainty across the projections of this magnitude of the change across different models by using performance constraining. We have modified the wording in text to reflect this more clearly (line 385).

Regarding the suggestion of using one model as a proxy for future observations in a perfect-model approach, we believe a test as proposed would answer the question of whether future variability depends on historical variability, specifically for each model across those considered. We agree that this would be an interesting exercise and an additional layer of rigour to the analysis proposed here. However, to answer the question of whether future variability depends on historical variability across different models we believe is at this stage beyond the scope of our current manuscript. To keep the analysis presented here as concise yet robust as possible, we have included an adapted version of this

perfect-model projections test suggested by the reviewer based on CESM2-LE alone, and find that indeed for most regions the constrained ensemble is closer to the chosen “good model”, as discussed above in more detail.

Is Fig. 6 mentioned in the text? Does Fig. 6 provide anything that's not in Figs. 4 and 5? Is a t-test the right test for a difference in variances? Or F-test? Though this is a difference in variances. Again, isn't comparing the full ensemble to one with fewer models a bit of a straw man? Also I cannot visually tell the difference between small values (white) and insignificant values (also white?). What does "general region" mean in the caption? Is there some smoothing?

Thanks for the comment - we have revisited and changed Figure 6 and this does not apply any more.

Line 267. "This evaluation enables end-users of these ensembles to select the best fitting model for their region and time of interest" Is there evidence that such a strategy leads to more accurate projections in the perfect model (each model in turn as observations) setting?

There is certainly evidence that this could allow studies in the historical period to be more accurate - even if evidence for future variability projections is more limited, as discussed above. Users could for example use a large ensemble to get extreme statistics in a model with realistic temperature variability for the historical period. This may not be possible in observations due to the low finite sample size.

Line 274. "they may, however, underestimate the projected temperature variability change" or they may not? Right?

Correct, this is why we used the word 'may' not do.

Line 277. "providing an improved estimate" In what sense is the estimate improved?

In this stance, we are referring to a potential reduction in uncertainty in future projections that results from constraining these projections to those models that exhibit adequate historical performance. We agree with the reviewer that the original wording was somewhat imprecise, and have updated this line to 'providing a performance-constrained estimate of both the projected change and the model uncertainty around it' for rigour.

Line 281. "In these regions we find that temperature variability itself is typically overestimated in present day climate; while the constrained ensemble provides a larger projected change in variability." It would be helpful to provide the reader with pointer to where these results appear. Or perhaps some summary graphic showing this relation?

We have added a reference to the figures that show these results in this sentence.

Line 310. "Our attempt to constrain projections of future temperature variability change is an example of change in the climate system that cannot yet be foreseen with the best available knowledge." What does this mean?

We are referring here to our findings indicating that, while constraining projections with historical model performance can reduce model uncertainties in some cases, these uncertainties are not completely erased. Even when successfully identifying several models that perform adequately in capturing today's climate in certain regions, model disagreement in temperature variability projections remains, in many cases, large. And thus, model differences across temperature variability change under future climatic conditions remain, over large areas, unreconciled. As we discuss above, remaining protection uncertainty across constrained ensembles of models that capture historical variability correctly could be driven by different climatic and environmental changes across different models. And these differences in future model behaviour in many cases (as the above mentioned example for the Amazon forest) cannot be reconciled using best available knowledge today, including evaluations such as those presented here. We have expanded this paragraph to reflect these aspects more clearly.

Line 365. "date" -> data

This has been changed as suggested.

Line 370. "Lastly, to allow a more finely resolved temporal analysis and to increase sample size we perform this evaluation on monthly mean temperature anomalies instead of seasonal averages." What evaluation? The rank-frequency evaluation or another diagnostic?

Thanks for bringing this to our attention. This statement refers only to the rank frequency evaluation. We have changed the line to read "Lastly, to allow a more finely resolved temporal analysis and to increase sample size we perform this rank-frequency evaluation on monthly mean temperature anomalies instead of seasonal averages." for clarity.

Line 384 "To illustrate how internal variability may affect rank frequencies" isn't the rank histogram itself assessing variability? Or do you mean finite sample size?

Yes, we are referring to the effect of insufficient variability sampling in the finite sample size provided by the relatively short observational record. This effect may cause the rank variability to be undersampled as well, resulting in a non-flat rank histogram, either for adequately performing models or even in a perfect-model framework. We have expanded on these issues on point 2. of the response to the reviewers remark above, and have clarified this point in the revised manuscript where relevant. This particular line has now been removed from the revised manuscript, but the wording has been updated to reflect this suggestion throughout the manuscript.

Line 397 "To quantify when variability biases are present, we select the following thresholds." These seem arbitrary. It would be sensible to include some sort of significance testing of rank histograms.

We thank the reviewer again for this suggestion. We have now expanded our evaluation framework to include an additional criteria reflecting the whole spectrum of ranks based on the perfect-model rank histogram range, as discussed in the response to point 2. above. We agree that these thresholds and also the percentile ranges are somewhat arbitrary, and that assessing rank frequencies across all ranks against

the perfect-model range is the superior evaluation benchmark. However, the complexity and computational costs of performing such an evaluation at the grid-cell level would be substantial, and would complicate the applicability and reproducibility of this evaluation framework by potential end-users. For these reasons we have decided to include the superior evaluation of the full rank histogram against the perfect model range for spatially averaged metrics as a primary criteria, and have maintained a revised version of this threshold-based criteria as a secondary condition in our evaluation, to ensure that a grid-cell performance assessment that is computationally sensible is also included in our framework. Also, although we agree that the thresholds chosen are somewhat arbitrary, they agree well with the ranges found in the perfect-model rank range evaluation for the assessed rank windows.

Line 408. "Note that the variability underestimation threshold of 8% is chosen to be rather conservative to account for the different ensemble size" Wouldn't it be more objective to use a standard significance test which accounts for ensemble size? Perhaps these are all points where a method that takes into account sample and ensemble size would be beneficial.

We again agree with the reviewer and have expanded our evaluation framework to reflect this, using the previously described perfect-model rank range which indeed reflects ensemble size as well as other model- and region-specific characteristics, as discussed in our response to point 2. above. For this additional Criteria 1., allowed frequencies for observations occurring outside of the ensemble spread as rank 0 or rank n (with n being the number of members) now emerge from the perfect-model frequency range, which by design takes into consideration both the ensemble size as well as the simulated regional temperature distributions. Upper bounds of this range go from around 2% (e.g., for CESM2-LE region L10) to above 10% (e.g., for GFDL-ES2M2 region L2; these values can be found in the Supporting Information figures). As discussed above, we have decided to also keep the fixed threshold assessment as additional criteria to incorporate a sensible and non-overtly complex grid-cell level assessment in our evaluation. We also maintain the discussion of the caveats of applying such fixed thresholds to evaluate SMILEs of different sizes included in our original manuscript.

Line 385 "and" -> an

This has been changed as suggested.

Line 428 "Lastly, we determine, per grid-cell, if a model simulates observed temperatures adequately when none of these biases are present." Is this saying when none of the biases in the previous paragraph are present, an additional procedure is applied to determine if a model simulation is adequate? If so what is that procedure?

We thank the reviewer for bringing this unclear statement to our attention. Indeed we are referring to the previously discussed biases, with no additional procedure incorporated in this stage. We have clarified this section to specifically name the two biases assessed, and this line to read 'Lastly, when neither bias a) nor b) are present, we determine a grid-cell to be unbiased.' for clarity

Lines 430–433 are a repeat of lines 111–114? I still don't quite get what the time series refer to.

We have added a more elaborated description of what the time series represent and how they illustrate the biases highlighted by our rank-frequency evaluation both in the results and methods sections, as described above.

Line 459. "Significance for Figure 7 is calculated using a t-test approach for each domain." Fig. 7 is the domains?

This is no longer relevant as while delving into the reviewer's comments we have found the t-test approach was not appropriate for our analysis.

Reviewer #2 (Remarks to the Author):

Review for "Temperature variability projections remain uncertain after constraining them to best performing SMILEs"

I have read the manuscript and have several concerns, mainly about the scope of the study. The authors essentially use output from large ensembles of climate model simulations, evaluate whether or not they have a good representation of historical temperature variability, then assemble a "constrained ensemble" based on admittedly subjective criteria to evaluate how much removing models with poor representations of historical variability changes future projections. In general, they find that the answer is basically unchanged with or without their constraint.

While this is a worthy effort, I do not believe this constitutes a novel study that adds to our understanding of temperature variability in climate model simulations. The authors provide a detailed description of their results, and while I have no objection to their findings or methodology (though I am slightly confused by the ranking figure) I do not believe this paper constitutes a sufficient advance to be considered in Nature Communications.

We thank the reviewer for their time and comments on this paper. We disagree with the reviewer that this is not a '*novel study that adds to our understanding of temperature variability in climate model simulations*'. We believe this study is novel for the following reasons:

1. Without evaluation studies such as this one we cannot know whether our models correctly simulate the world we live in. We cannot improve our models, nor effectively analyse their results without this knowledge. Suarez-Gutierrez et al, 2021 was the first paper to be able to investigate model performance of temperature variability by utilizing SMILEs. This paper, published less than 5 years ago, already has 84 citations, highlighting the value of such work. This work has been cited extensively both by users applying the novel model evaluation approaches introduced there within the climate science community, as well as by users both within and outside this community (e.g. climate model data end-users across risk assessment, economic, agricultural

and environmental modelling communities, etc.) aiming to assess and compare the performance of their climate model(s) of choice without the need to carry out extensive analysis themselves. Therefore, we believe bringing accessible and robust multi-model evaluation results, particularly for impact-relevant metrics such as temperature variability, is crucial and will be of great benefit both for the climate modelling community and for its end-users, and therefore be of great interest to the Nature Communications readership.

2. This work expands on previous related work such as Suarez-Gutierrez et al, 2021, first, by expanding the evaluation framework used to incorporate more robust and formally defined evaluation criteria, and second, by considering and evaluating metrics more closely related to climate-driven environmental and socioeconomic impacts. This work offers the first evaluation of all available CMIP5 and CMIP6 SMILEs for temperature variability in local summer and winter seasons. Evaluating such variability is crucial for the representation of climate extremes, as well as for example for ecosystem stability or agricultural productivity, in a way that most commonly assessed metrics such as mean temperature or annual temperature variability are not. As such, our present study builds on previous work and provides an essential evaluation of how well state-of-the-art climate models represent temperature variability. Furthermore, following the suggestions of Reviewer 1, we have also modified our evaluation framework to incorporate formally defined and more objective evaluation criteria.
3. Building on points 1. and 2., this evaluation will provide a basis for model selection and assessments for the evaluation of temperature related extremes, which cannot be correctly simulated if the underlying variability is incorrect. Therefore our manuscript will provide an essential tool and guideline for future research on these issues.
4. The latest IPCC assessment report flagged future projections of temperature variability as uncertain, with large model disagreement. The goal of our manuscript is to determine whether this is due to incorrect representation of temperature variability today, and whether uncertainty can be reduced by introducing performance constraining. A possible null hypothesis to explore is that future temperature variability depends on historical variability, and therefore, by selecting models that best match observations and agree on their current variability representation, future projections across best-performing models will also agree more and thus be less uncertain. In contrast, the alternative hypothesis is that future temperature variability is intrinsic to how the Earth system evolves over different regions, and therefore even constraining projections of future temperature variability change to models that best capture historical variability will not necessarily reduce future uncertainty if those models that capture historical variability adequate still exhibit diverging behaviours in the future. To the best of our knowledge, our findings are the first to reveal that, for large areas of the world, the second hypothesis is true. This means that, first, uncertainty in temperature variability change cannot be fully constrained nor reduced globally based on historical model performance. Second, the tendency in the climate modelling community towards pooling resources and developing fewer, less independent models, may lead to an insufficient sampling of model uncertainty, at least for this particular metric of

temperature variability, which is an example of change in the climate system that cannot yet be foreseen with the best available knowledge.

5. Lastly, while our results indicating that variability change projections remain largely unchanged by picking the models that perform well historically may seem uninteresting, to the climate modelling and evaluation community this highlights a critical knowledge gap; an area of uncertainty in our future projections that cannot just be improved by picking ‘the good models’. This underscores the need for further investigation, model development and improvement, and the use of multiple models to robustly estimate the range of potential changes for this climate- and impact-relevant metric. Results such as those presented here provide strong arguments against moving towards one ‘super climate model’ and will hopefully guide future modelling efforts to be as efficient and rigorous as possible.

To highlight and reflect upon these issues in the manuscript, we extensively expanded on the introduction and discussion sections, and have added the following conclusions and implications (starting in line 458):

“(i) We provide a comprehensive evaluation of the state-of-the-art SMILEs ability to represent the historical summer and winter temperature variability in observations. We identify CESM-LE and CESM2-LE as the SMILEs that provide the best representation of isolated temperature variability as well as of both temperature variability and forced change, with GFDL-SPEAR-MED and MPI-GE5 as close third in each category respectively. This multi-model evaluation across all available CMIP and CMIP6 SMILEs provides a basis for model selection for the assessment of temperature variability and extremes, which cannot be correctly simulated if the underlying variability is incorrect.

(ii) Our evaluation also shows that some regions are systematically not well represented such as the Southern Ocean (O9), the Indian Peninsula (L20), and Northern Australia (L22), as well as the Amazon basin (L6), Northern (L12) and Eastern Africa (L14), and Southern Australia (L23), all specifically in the local summer season. We also find that certain areas of the world are systematically well captured by most models, especially in the local winter season, namely North America (L1-L4), Southern South America (L8), Europe and Northern Africa (L9-L11) and North and Central Asia (L16-L19). Finally, we conclude that local winter seasons are generally simulated better by most models more often than local summer seasons.

(iii) Similar to the IPCC, we find low model agreement on the sign of the change over large parts of the Earth’s surface and large model disagreement on the magnitude of the projected change under future warming, particularly over the land surface and the tropical Pacific Ocean. This implies that we cannot afford to move away from multi-model ensembles that sufficiently capture the uncertainty in our future projections.

(iv) The constrained ensemble decreases model spread over some regions and gives a lower range of projected futures, providing a smaller range of potential future projections and greater model agreement, particularly in South America, India, Australia & South-East Asia, the west Pacific Ocean and Southern

Ocean in DJF and north South America, Africa, parts of Australia and the Southern Ocean in JJA. However, the constraint does not substantially increase model agreement on the sign of the projected change.”

Major Comments:

Figure 1: The shading is extremely difficult to see, which makes this plot difficult to interpret. I found the discussion of rank histograms slightly confusing and thought the authors should say more about what a “desirable” scenario would be for these histograms. Presumably it would be that the observations sit somewhere in the middle of the ensemble, but that’s just my guess.

We thank the reviewer for bringing these issues to our attention. We have modified the format of the figure to improve its readability, which should be corrected now in standard non-web based pdf readers. We have also expanded the discussion along this figure (e.g., lines 114-197) as well as in the methods section (lines 514-625) to clarify our evaluation framework, including what desirable performance outcomes may look like (lines 163-168), as well as what the different biases considered may look like for the updated evaluation criteria used (lines 177-186).

The discussion of detrended versus non-detrended data was interesting, but qualitative and speculative. I was also confused by lines 172-175, where the authors say “The higher agreement between non-detrended data and observations could arise from the fact that misrepresented forcing signals can counteract commonly-found variability overestimation biases” Wouldn’t this be the opposite? I would think that higher agreement between non-detrended data and observations would arise from accurate representation of the forcing signal, rather than misrepresentation.

We have expanded the results section regarding the comparison of detrended and non-detrended evaluation results, and we also include an example of these two versions of our evaluation in Figure 1 and related discussion. We provide a quantitative comparison on model performance based on regions that are captured adequately for detrended and non-detrended data as well as potential arguments exploring why performance may be affected by detrending in the results section (lines 177-186, 198-210, 220-280).

In regards to the comment on ‘higher agreement between non-detrended data and observations could arise from the fact that misrepresented forcing signals can counteract commonly-found variability overestimation biases’, we thank the reviewer for highlighting this unclear point. In general, for detrended data, good agreement between models and observations indicates that models capture temperature variability correctly. When evaluating non-detrended data, good agreement indicates that models capture both variability and forced signals correctly. Therefore, since the second is a stricter test that requires both the variability and the forced signal to be well captured, models would perform either equally well (for the cases where the variability was already well captured and the now non-detrended forced signal is also well captured) or worse (for the cases where the variability was already well captured and the now non-detrended forced signal is not well captured). Generally, the cases where the forced signal is well captured but the variability is not would still yield a negative performance evaluation with non-detrended data. Therefore, in theory, we do not expect improved performance by assessing non-detrended data.

However, we find limited examples of cases where our evaluation indeed yields improved agreement using non-detrended data, as we noted in the line highlighted by the reviewer. What we mean in this case is, rather than revealing an improved agreement, the use of non-detrended data is confounding our evaluation and masking biases. For this, we refer to an example for this situation, which has also been added to the revised manuscript (see Figure 1 L12 panel and Fig. D below).

We see from the non-detrended evaluation for region L12 simulated by CanESM5 (left panels in Fig. B), that observations occur slightly towards the upper portion of the ensemble in the early 20th century, being somewhat warmer than most simulations. However, towards the end of the record the forced warming signal in the model appears to be stronger than observed, and observations clearly cluster in the bottom section of the ensemble, while in both cases remaining well within the ensemble limits. This results in a relatively flat rank histogram (color line in left rank histogram in Fig. D) that is within the perfect-model rank histogram range (gray dashes lines in left rank histogram in Fig. D), and therefore classified as an adequate representation in our non-detrended evaluation.

In contrast, the evaluation of detrended data removes the bias in the forced warming and reveals a variability overestimation bias, with observations occurring too frequently along the central ranks of this ensemble (right panels in Fig. B). This results in the observations' rank frequencies that are larger than the perfect-model rank range for the central rank windows, and therefore classified and inadequately represented in our detrended evaluation.

Due to this too large variability and too strong warming signal in the model, historical non-detrended observations cross the ensemble without exceeding the ensemble limits with a frequency that appears uniform when assessing the non-detrended rank histogram, but is revealed to be indeed biased upon closer inspection for the detrended evaluation. Since we use exclusively detrended data for our performance constraint, we expect such confounding factors to not affect our evaluation. However, we still want to bring these issues to the readers attention, since potential end-users may choose to select models based on their non-detrended performance. We have clarified and expanded on these aspects in the revised manuscript (lines 267-278).

Figure D: Example of the rank-frequency evaluation framework for JJA temperatures in CanESM5 against GISTEMPv4 observations. Time series and rank histograms show the evaluation of spatially averaged temperatures for region L12 (Evaluation Criteria 1). Time series show the ensemble maximum and minimum (color lines) and central 75th percentile ensemble spread (shading) against observations (dots). Rank histograms show observations rank frequency aggregated over a 3-rank bin (bars), the running mean rank frequency over a centered $n/5$ rank window (lines; for 1 to $n-1$ ranks) and the absolute frequencies of rank

0 and n (crosses), with n the number of ensemble members, for observations (color) and perfect model rank range (gray). If observations rank frequency is within perfect-model range for all rank windows, Criteria 1 is met and this is highlighted by a green star at the top right; if not, by a red cross. Percentages at the top left show the frequency of regionally averaged observations occurring above (red) or below (blue) ensemble limits, or clustering within the central 75th percentile range (gray), analogous to the grid-cell evaluation in Criteria 2.

Minor Comments: Figure 6 never called or discussed?

We agree, this figure was not particularly useful and has been removed and replaced with a new Figure 6 and 7. We now include Figure 6 that compares temperature variability in the models at 1degree of warming with the observational record. Figure 7 has been updated to consider temperature variability at 3 degrees over regions that are poorly represented in climate models. This new Figure compares the constrained and full ensembles and the constrained ensemble with the results from CESM2, which we found to be one of the best performing models over these regions. Below is the discussion on this new Figure found on line 353.

“While we cannot assess whether the future projections are more realistic in the constrained ensemble compared to the full ensemble, as we do not have observations of the future, we can compare the projections with those from the most realistic model over these poorly modelled regions in the historical period. CESM2 is one of the two models that best represent Australia & South-East Asia (L21-23), South America (L6-8), and Africa (L12-16; the other being CESM-LE), with 15 “good performances” out of 22 across both regions and seasons. By using this model as a proxy for possible future observations, we can compare the differences at 3 °C of warming between the full and constrained ensemble, the constrained ensemble (excluding CESM2), and CESM2 (Fig. 7, see Figs. 10 and 11 for a spatially aggregated comparison across all regions). We find that, except for South America (where CESM2-LE fails to capture historical variability according to our evaluation for L6 and L7) and Africa in DJF the constrained estimate is closer to CESM2 (our “good model”) than the full ensemble, tentatively suggesting that our future projections might be more realistic in the constrained ensemble compared to the full ensemble. ”

Response to the Reviewers' comments on manuscript NCOMMS-25-1462A "Temperature variability projections remain uncertain after constraining them to best performing SMILEs" by Suarez-Gutierrez & Maher.

Please find enclosed our response to the Reviewers' remarks on our manuscript.

We thank the reviewers for their assessment and remaining comments, which we address individually more in depth (in blue).

Reviewer #1 (Remarks to the Author):

The authors have responded carefully to all my comments. I have one comment about a possible significance testing approach and a few minor comments.

The authors state in their response "Therefore, if the reviewer has any specific suggestions as to how to appropriately calculate significance for this study we are open to including them." I think that a permutation test is appropriate in this case, as it does not rely on distributional assumptions and is exact under the null hypothesis of exchangeability. Specifically, compute the difference in variances between the constrained and full ensembles. Then, repeatedly create random constrained ensembles (of the same size etc) by sampling without replacement from the full ensemble, and compute the same variance difference for each permutation. The p-value (two sided) is the fraction of permutations where the absolute value of the permuted variance difference exceeds that of the observed difference. I think that your calculation mentioned at line 380 is very close to this.

We had actually considered doing this bootstrap type in response to the last revision, but decided against it as we believe the test is not entirely appropriate. The suggested calculation would tell us whether the constrained sample is different to a random subsample of the same number of members as we have constrained to. However, the actual question we believe needs to be answered is whether the constrained estimate itself is statistically different from the full ensemble estimate. We believe this cannot be assessed with the suggested bootstrapping approach and, therefore, we argue against including it. We thank the reviewer again however for this suggestion, and in general for their dedicated response.

Minor comments

Line 46 define SMILE at first use in text (not at line 83)

Thanks for catching this - we have fixed the issue.

Line 56 "with 60N"?

We have changed this to read "with 60N the latitude where the largest changes in fractional snow cover occur".

Line 57 "This is also a region of decreasing variability" what is "this"?

We have changed this to read "The 60N latitude is also a region..." to clarify this.

Line 72 "Therefore, a possible hypothesis is that future temperature variability depends on historical variability" I don't know what this means. I think you want to mention models. Perhaps, "Therefore, a possible hypothesis is that the ability of models to project future temperature variability depends on their ability to simulate historical variability"

Thanks for the suggestion - we have changed this as suggested.

Line 78 "In contrast, the alternative hypothesis is that future temperature variability is intrinsic to how the Earth system evolves over different regions." I don't know what this means.

We have reformulated this sentence for clarity and now reads "In contrast, the alternative hypothesis is that different models' projections of future temperature variability depend not simply on past temperature variability, but rather on how the Earth system as a whole evolves over different regions in each model. If this were true, subsampling models based on their historical performance would not necessarily reduce future uncertainty, as long as those historically best-performing models still exhibit diverging behaviours in their representation of future temperature variability changes."

Line 128 Delete comma

Done as suggested.

Line 153 Readers will recognize "perfectly flat" as a bit of a straw man since no tests require perfect flatness, and the usual tests take sample size into account. The usual chi-squared test does (I think) assume independence. You could drop the sentence starting at line 150 and simply say at line 154 "We tested the flatness of the rank histogram using a perfect-model rank range test which make no distributional assumptions and takes into account both ensemble size and serial correlation"

We thank the reviewer for bringing this to our attention. We have rephrased this section for clarity, which now reads:

"However, the fact that we are evaluating free-running uninitialised simulations against a relatively short observational record means that the internal variability in the climate system may be insufficiently sampled on these time scales, and thus rank histograms may appear to be not flat, even for perfectly performing models.

To overcome this, we assess the flatness of the rank histogram using a perfect-model rank range test, which does not make any assumptions about the shape of the underlying distribution and takes into account both ensemble size and serial correlation."

Line 166 and throughout "Criteria" is plural so Criterion 1?

We thank the reviewer for the suggestion, although we believe Criteria is also appropriate as it is very commonly used in singular case, and we would prefer to keep it as is.

Line 254 space before GFDL
Done as suggested.

Lines 337 and 338 em dashes instead of hyphens
This has been changed to commas.

Reviewer #2 (Remarks to the Author):

In my original review, I expressed concerns about the novelty of this study and the importance of the results. The authors have done a better job at explaining why their study is different from the original 2021 study that used a similar approach to understand temperature variability in the CMIP5 models. I appreciate the new description of the two evaluation criterion, which helps clarify what distinguishes this approach from the previous efforts to understand temperature variations in climate models.

The result that the constrained ensemble does not help projections of temperature variability converge is an interesting one, but in the authors' response they write a hypothesis that "future temperature variability is intrinsic to how the Earth system evolves over different regions, and therefore even constraining projections of future temperature variability change to models that best capture historical variability will not necessarily reduce future uncertainty if those models that capture historical variability adequately still exhibit diverging behaviors in the future." This is definitely supported by their results, but without an explanation of why the models disagree with respect to future climate change, it's not really satisfying to me as a result. I completely understand, and agree with, the authors' point that "uncertainty in our future projections that cannot just be improved by picking 'the good models'" but I think this isn't really a knowledge gap, but more of a philosophical problem in the climate modeling community.

In general the authors have addressed my comments thoroughly, which I appreciate given my somewhat critical review.

We thank the reviewer for their insights. Regarding their point about reducing uncertainty in future projections, we agree with part of the issue being a philosophical question, yet the question is indeed in practice often being answered by focusing on fewer, potentially 'better' or at least more complex models. Therefore, we believe presenting perspective as those in this paper to the modeling community is imperative to ensure that both reducible and irreducible projection uncertainties are addressed as robustly as possible. We want to thank the reviewer again for their time and feedback.